# Spatial Variations in Silicate-to-Nitrate Ratios in the Southern Ocean Surface Waters are Controlled in the Short Term by Physics Rather Than Biology

Pieter Demuynck[1], Toby Tyrrell[1], Alberto Naveira Garabato[1], Mark C. Moore[1], and Adrian P. Martin[2]

[1]Ocean and Earth Science, University of Southampton, Southampton SO14 3ZH, UK
[2]National Oceanography Centre, Southampton SO14 3ZH, UK

**Correspondence:** Toby Tyrrell (toby.tyrrell@soton.ac.uk)

**Abstract.** The nutrient composition (high in nitrate but low in silicate) of Subantarctic Mode Water (SAMW) forces diatom scarcity across much of the global surface ocean. This is because diatoms cannot grow without silicate. After formation and downwelling at the Southern Ocean's northern edge, SAMW re-emerges into the surface layers of the mid- and low-latitude oceans, providing a major nutrient source to primary producers in those regions. The distinctive nutrient composition of SAMW originates in the surface waters of the Southern Ocean, from which SAMW is formed. These waters are observed to transition from being rich in both silicate and nitrate in high-latitude areas of the Southern Ocean, to being nitrate-rich but silicate-depleted in SAMW formation sites further north. Here we investigate the key controls of this change in nutrient composition with an idealised model, consisting of a chain of boxes linked by a residual (Ekman- and eddy-induced) overturning circulation. Biological processes are modelled on the basis of seasonal plankton bloom dynamics, and physical processes are modelled using a synthesis of outputs from the data-assimilative Southern Ocean State Estimate. Thus, as surface water flows northward across the Southern Ocean toward sites of SAMW formation, it is exposed in the model (as in reality) to seasonal cycles of both biology and physics. Our results challenge previous characterisations of the abrupt northward reduction in silicate-to-nitrate ratios in Southern Ocean surface waters as being predominantly driven by biological processes. Instead, our model indicates that, over shorter timescales (years to decades), physical processes connecting the deep and surface waters of the Southern Ocean (i.e. upwelling and entrainment) exert the primary control on the spatial distribution of surface nutrient ratios.

## 1 Introduction

The Southern Ocean (SO) is an important component of the Earth system in its own right, but also through the influence it exerts over a large fraction of the rest of the ocean through nutrient supply. It was hypothesised (Sarmiento et al., 2004), and

is now generally accepted, that Subantarctic Mode Water (SAMW, see Figure 2) acts as a conduit carrying nutrients from the SO to the mid- and low-latitude oceans, thus controlling the productivity of those regions. SAMW flows along the global ocean thermocline (at depths of 100 - 500 m, potential density anomaly $\sigma_\theta$ of 26.8 $kg\,m^{-3}$), and supplies the surface layers with nutrients via upwelling centers such as e.g., equatorial Pacific and off South America (Sarmiento et al., 2004). Global ocean model runs in which SAMW is artificially altered to contain no nutrients lead to up to a four-fold reduction in primary production outside the SO (Sarmiento et al., 2004; Palter et al., 2010). A range of biogeochemical processes in the upper limb (Fig. 2) of the SO overturning circulation modify the water properties of surface waters subducting in the SAMW formation sites. Properties acquired by these water during their time at the surface in the SO thus exert an important influence on the biogeochemistry of many upwelling regions elsewhere in the global ocean.

An important feature of the SO overturning's upper limb is the meridional gradient of surface nutrient concentrations, with highest values in the south and a northward reduction in nutrient levels. The gradient is most substantial for silicate: from more than 50 $mmol\,m^{-3}$ at the high-latitude winter-ice boundary, to 10 $mmol\,m^{-3}$ and less at the Polar Front, according to observations (Tréguer and Jacques, 1992). A similar decline was observed along a 42°E section, with marked steps in nutrient concentrations at each SO front (Pollard et al., 2002). Assmy et al. (2013) reported decreases of Si from 70 $mmol\,m^{-3}$ in the upwelling waters to less than 5 $mmol\,m^{-3}$ north of the Polar Front. Along that same transect, nitrate concentrations only decreased from about 30 $mmol\,m^{-3}$ to 23 $mmol\,m^{-3}$. In the Southern Ocean component of the United States Southern Ocean Joint Global Ocean Flux Study (JGOFS), AESOPS (Antarctic Environment and Southern Ocean Process Study) performed several cruises along a transect at 170°W. During those cruises (in Oct. 1997, Nov. 1997, Dec. 1997, Jan. 1998 and Feb. 1998) significant gradients in macronutrients were observed: nitrate concentration of about 25 - 30 $mmol\,m^{-3}$ at 68°S to 15 $mmol\,m^{-3}$ at 50°S and silicate concentrations decreased from 60 - 70 $mmol\,m^{-3}$ at 68°S to 0-10 $mmol\,m^{-3}$ at 50°S (Smith et al., 2000). Fig. 1 shows surface nitrate and silicate as observed during a German SO JGOFS cruise along a section at 5°E (Read et al., 2002). The gradients and the difference between silicate and nitrate are clear. This feature, present in all sectors of the SO, has important consequences for ocean biogeochemistry on a global scale. As a result of the residual surface nitrate in mode water formation areas, a considerable amount of nitrate is subducted into SAMW and Antarctic Intermediate Water. The depleted levels of silicate in the subduction regions leads to a relatively modest amount of silicate being exported from the SO, hence preventing diatom blooms in nitrate-rich but silicate-poor regions of the global ocean (Assmy et al., 2013).

The central question addressed in this paper is why the surface waters that subduct to form SAMW are depleted in silicate but not nitrate. It is not immediately obvious why this should be. According to the simplified view shown in Fig. 2, Circumpolar Deep Water (CDW) upwells at the Southern Boundary of the Antarctic Circumpolar Current (ACC) to provide surface closure of the SO overturning's upper limb. CDW has a higher concentration of silicate than nitrate (50 - 70 and 20-30 $mmol\,m^{-3}$, respectively), as do surface waters near the Southern Boundary. Diatoms at a variety of locations around the world typically take up silicate and nitrate in a ratio close to 1:1 (Brzezinski, 1985), and therefore it might be expected that, if removal by diatoms is the primary biogeochemical process in operation, that nitrate would run out before silicate rather than the opposite.

The preferential removal of silicate to nitrate in SO surface waters is generally attributed to biological processes, for two reasons. First, it has long been known that diatoms in the SO are hypersilicified. The diatoms seen in the SO are observed to

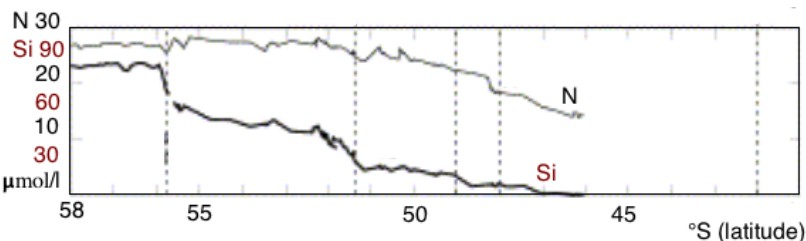

**Figure 1.** Nitrate and silicate concentration in the surface layer observed during a German SO JGOFS cruise along 5°E, Figure modified from Pollard et al. (2002)

have unusually thick frustules, and their Si:N ratios often greatly exceed the average of 1:1 (Brzezinski et al., 2002). This could be a result of physiological acclimation to the iron limitation that prevails in the SO. A culture experiment by Timmermans et al. (2004) found SO diatoms to become hypersilicified in the absence of iron. They found a clear correlation between iron concentration and silicate consumption with increasing Si:N ratios under more deplete iron concentrations. It could also be

a result of selection between species: the diatom species in the SO are more silicified than the average diatom species even under iron-replete conditions. For example, the ratios recorded by Timmermans et al. (2004) during their experiment were higher than the 1:1 average even under iron-replete conditions: 2:1 for the species *Actinocyclus sp.* and 3:1 for the species *Thalassiosira sp.* (increasing to 5:1 and up to 18:1, respectively, under iron-deplete conditions). Even for non-hypersilicified species, iron stress tends to increase the Si:N ratio to values higher than 1:1 (Assmy et al., 2013). Second, N and Si export

are thought to be uncoupled (Tréguer and Jacques, 1992; Pollard et al., 2002; Assmy et al., 2013), in the sense that particulate Si (diatom frustules) sinks relatively fast and either dissolves at great depth or does not dissolve, ending up in the siliceous ooze (Assmy et al., 2013). A modelling study by Holzer et al. (2014) showed that the average phosphate regeneration depth is $\sim 600$ m, whereas the corresponding mean depth of silicate regeneration in the SO is $\sim 2300$ m. Silicate that remineralises in the CDW is transported back southwards at depth, and re-surfaces near the Southern Boundary. Thus, silicate ends up in a

vertical recycling loop, and becomes efficiently trapped. Further, remineralisation of nitrate and silicate is different, as bacteria are very efficient at decomposing organic nitrogen in relatively shallow layers. This results in a significant proportion of the nitrogen demand being provided by efficient bacterial recycling of organic nitrogen, leading, on occasion, to mass sinking of empty diatom frustules with no organic matter left inside them (Assmy et al., 2013). Almost half of the global Si inventory goes through a SO-to-SO loop (Holzer et al., 2014). While phosphate (similar to nitrate) last used in the SO has only a 56%

probability of reemerging in the SO photic zone and of being used in the SO again, the probability for SO silicate being used in the SO again is as high as 95% (Holzer et al., 2014).

     In this paper, we assess the controls on the meridional gradient in Si-to-N ratios characteristic of the SO with an idealised model representing all of the region's important physical and biological processes. Our results highlight the pivotal role of physical processes in sustaining the nutrient distribution, on timescales up to decadal.

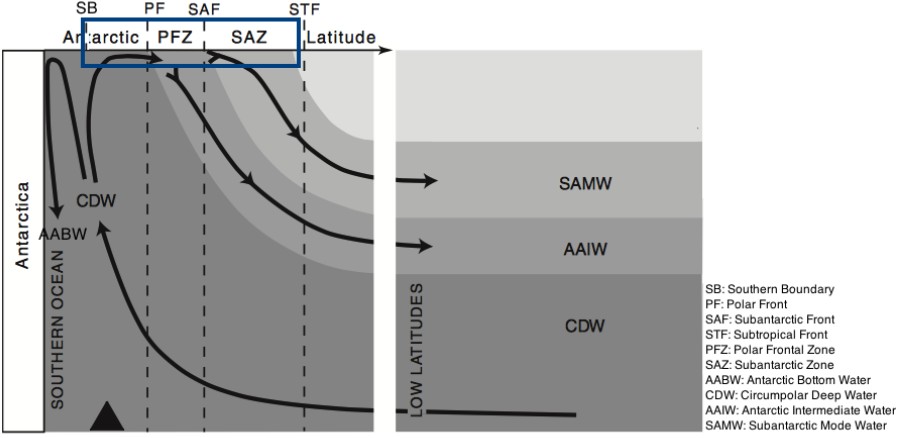

**Figure 2.** Circulation of main water masses in the Southern Ocean, Figure modified from Anderson et al. (2009). The box marks the region of interest

## 2    Model Description

In many studies, simple box models without vertical resolution have been used to study biogeochemical cycles at a certain location over a certain period of time (e.g. Tyrrell and Taylor (1996); Taylor et al. (1991)). In other studies 1-D box models with a vertical resolution have been used as well (e.g. Pondaven et al. (2000)). The advantage and strength of these models is
their simplicity and robustness. However, they lack spatial (zonal and/or meridional) dimension. Our model needs to cover a meridional range from the Southern Boundary to the downwelling zones at the Southern Ocean's northern edge. The modelling approach in this work consists in using a simple box model at pre-defined latitudes from the Southern Boundary to the northern edge of the SO and linking the boxes with an appropriate physical scheme as shown in Fig. 3.

### 2.1    Physical model

The vertical partitioning of the model consists of two active model layers and is based on seasonal changes in water mass properties as observed along a section in Drake passage (Evans et al., 2014). In winter, strong surface cooling and wind-driven mixing between the deepening surface layer and the underlying Antarctic Winter Water (AAWW) form a thick AAWW layer extending up to the surface. During summer, the upper part of AAWW is eroded by surface warming and internal mixing, and AAWW becomes a layer below a relatively shallow surface layer. The basic building block of the model is hence a 1-D
box model in which the water column is divided into a mixed layer (ML) and a subsurface layer (SSL). These two layers correspond to the observed surface ML and the AAWW. The depth at which the boundary condition is imposed at a certain latitude is defined as the thickest ML taken from a data-constrained estimate, as described in section 3.2) rounded up to the nearest 100 m (See Table 1). The lower boundary of the SSL is fixed in depth at each latitude. Model boundary conditions are

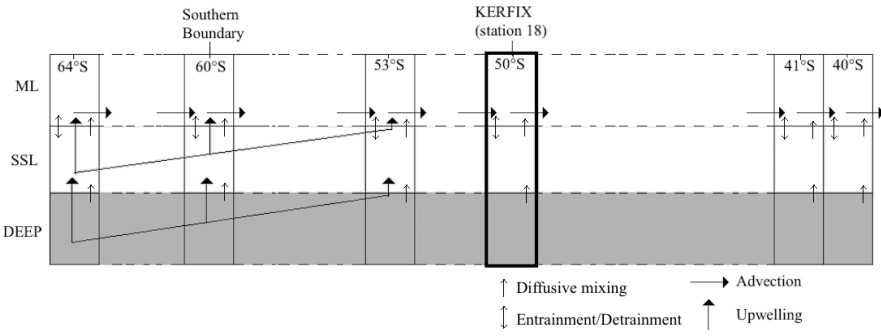

**Figure 3.** Structure of physical model: 1 box model contains a ML and a SSL. Northward advection in the ML connects the boxes in the meridional direction

**Table 1.** Lower boundary of SSL per station in m below water surface

| Station | 1 | 2 | 3 | 4 | 5 | 6 | 7 | 8 | 9 | 10 |
|---|---|---|---|---|---|---|---|---|---|---|
| Depth [m] | 200 | 200 | 300 | 400 | 400 | 400 | 400 | 400 | 400 | 400 |
| Latitude [°S] | 63.52 | 62.92 | 62.31 | 61.68 | 61.04 | 60.39 | 59.73 | 59.05 | 58.35 | 57.65 |

| Station | 11 | 12 | 13 | 14 | 15 | 16 | 17 | 18 | 19 | 20 |
|---|---|---|---|---|---|---|---|---|---|---|
| Depth [m] | 400 | 400 | 400 | 300 | 300 | 300 | 300 | 300 | 300 | 300 |
| Latitude [°S] | 56.93 | 56.19 | 55.44 | 54.68 | 53.90 | 53.11 | 52.30 | 51.48 | 50.64 | 49.79 |

| Station | 21 | 22 | 23 | 24 | 25 | 26 | 27 | 28 | 29 | 30 |
|---|---|---|---|---|---|---|---|---|---|---|
| Depth [m] | 300 | 500 | 500 | 500 | 500 | 200 | 200 | 200 | 200 | 200 |
| Latitude [°S] | 48.92 | 48.03 | 47.13 | 46.22 | 45.29 | 44.34 | 43.38 | 42.41 | 41.41 | 40.41 |

applied at this boundary, representing properties (assumed constant over time) of deep water beneath the SSL. In the model, the depth of the SSL is calculated as the difference between the depth of the fixed boundary and the ML depth. In summer, the ML is thin and the SSL is relatively thick, and vice versa in winter.

Starting at the Southern Boundary ($\sim$60°S) surface waters will move northward with a characteristic velocity of order 0.3 - 0.4 $km\,d^{-1}$, and eastward with a characteristic velocity of order 20 $km\,d^{-1}$ based on the routes of six CARIOCA drifters in an area in the subantarctic zone (38°S - 55°S, 60°W - 60°E) from January 2006 to April 2008 (Merlivat et al., 2015)). In our work, the emphasis is on the northward flow of the water and processes affecting water properties following this flow. It is thus necessary for the model to span the region meridionally from the Southern Boundary ($\sim$60°S) to the latitude at which surface waters subduct near the Subtropical Front (somewhere between 45°S and 30°S depending on longitude). The meridional range included in the model is 65°S to 40°S. It is implemented by discretising the distance in that range. 30 stations (see Table 1) are defined between the northern and southern edge of the model. Each station is represented by a simple biogeochemical box model containing a SSL and a ML. While the intense eastward flow and the zonal connectivity of the SO would suggest

that zonal advection should be included in the model, we omit the zonal dimension of the circulation here to preserve model simplicity. Also, as explained more elaborately in section 3, the scarcity of data that we can use for boundary conditions forces us to use averaged data over the zonal dimension and of course we try to come to general conclusions for the SO which also justifies the omission of the zonal dimension.

Northward advection is induced by the strong and persistent wind (predominantly westerlies) over the Southern Ocean. In his summative paper, Deacon (1982) defines the Antarctic Divergence as the transition between easterlies and westerlies (situated between 62°S and 72°S). It is an Ekman divergence zone, in which circumpolar deep water reaches the surface and flows both north- and southwards. This upwelling is prevalent over an extensive latitudinal range. As such, our model should include both upwelling and northward advection over a substantial latitudinal band north of the Southern Boundary.

Interaction between the ML and the SSL at a station is represented via diffusive mixing. Mixing is defined here as the tendency to homogenise water properties exhibiting a spatial gradient. Advection (including upwelling) and diffusion of a chemical concentration C are mathematically described by the well-known advection-diffusion equation:

$$\frac{\partial C}{\partial t} = D \bigtriangledown^2 C - \boldsymbol{v} \cdot \bigtriangledown C \tag{1}$$

Written out, this becomes:

$$\frac{\partial C}{\partial t} = D_x \frac{\partial^2 C}{\partial x^2} + D_y \frac{\partial^2 C}{\partial y^2} + D_z \frac{\partial^2 C}{\partial z^2} - v_x \frac{\partial C}{\partial x} - v_y \frac{\partial C}{\partial y} - v_z \frac{\partial C}{\partial z} \tag{2}$$

The first three terms of the equation describe diffusion with D denoting the diffusion coefficient along a certain axis and assumed constant. The last three terms describe advection. It is not easy to determine the value of D. Generally, horizontal diffusivity is assumed to be larger than vertical diffusivity by a factor of $10^7$ (Garrett, 1979). The northward advective velocity $v_x$ is about 5000 times larger than the upwelling velocity $v_z$. This suggests that horizontal diffusion may be more important relative to horizontal advection than vertical diffusion relative to vertical advection. However, the tracer concentration gradient will be larger in the vertical direction than in the horizontal direction. We will assume that horizontal diffusion is of minor importance compared to horizontal advection, as the diffusive term entails the calculation of the second derivative of a variable which will be an order of magnitude smaller than a first derivative featuring in the advective term. Horizontal diffusion will thus be omitted. In turn, vertical diffusion will be included in the model in a simplified form that is often adopted in simple box models. Meridional advection and upwelling will also be included in the model. Equation 2 then becomes:

$$\frac{\partial C}{\partial t} = D_z \frac{\partial^2 C}{\partial z^2} - v_x \frac{\partial C}{\partial x} - v_z \frac{\partial C}{\partial z} \tag{3}$$

Physics, including advection, upwelling and entrainment/detrainment provide the link between the layers in the vertical direction and between the boxes in the meridional direction. In the model, upwelling is made to take place in the first 15 stations. $Q_{upw}$, the vertical transport of water, decreases from $8650\,m^3\,d^{-1}$ at 63.52°S (calculated as the product of the estimated upwelling velocity at that latitude (Morrison et al., 2015) with the horizontal area of the box) to zero at 53.11°S. Vertical transport from the SSL to the ML is the same as from the deep layer to the SSL. Conservation of mass enables the calculation of the horizontal transport into ($Q_{adv,in}$) and out of ($Q_{adv,out}$) the ML in a box i. Doing so, we find an average northward flow

in the model of about $46\,000\,m^3\,d^{-1}$ per m width. If we assume an average MLD of 100 m over all stations, we find a northward velocity of magnitude of order $0.46\,km\,d^{-1}$. This compares well with the values found by Merlivat et al. (2015). Note that horizontal transport is limited to the ML; there is no horizontal transport in the SSL. This is a model specific simplification and is partially justified by the fact that Ekman flows typically extend to depths of about 50 - 100 m (Talley et al., 2011). Note that this will not affect the mass balance because our starting base for advection is volume transport and not velocity of the water. Upwelling and advection of nutrients are therefore calculated using $-v_x \frac{\partial C}{\partial x} - v_z \frac{\partial C}{\partial z}$ (second part of Eq. 3).

Fig. 3 is a visualisition of the physical structure of the box models and how they are linked together via advection and upwelling. For biogeochemical and physical processes happening within one box model see Fig. 4.

In the box model a vertical gradient of a variable $\frac{\partial C}{\partial z}$ is established between the deep layer, the SSL and the ML. The diffusive flux of a variable C from a layer i+1 to the layer i above $\left( D_z \frac{\partial^2 C}{\partial z^2} \right)$ is replaced by:

$$F_{diff} = \frac{k_{mix}}{h}\left(C_{i+1} - C_i\right) \tag{4}$$

where $h$ is the ML or the SSL thickness. $k_{mix}$ is the mixing coefficient set to $0.1\,m\,d^{-1}$.

Each model variable is subject to entrainment and detrainment. Entrainment is a process changing the concentration of a tracer due to an increase in the ML depth (this is called winter mixing in the model study by Pondaven et al. (2000), but for the sake of clarity, it is termed entrainment here). If the ML deepens in winter, water with a different concentration enters the ML. Note that entrainment only occurs when the ML deepens. When the ML shoals, the concentration of a tracer within the ML remains unchanged, but the concentration in the SSL changes. This process is called detrainment. Entrainment changes the concentration of a variable C as:

$$C_{ML+} = \frac{\Delta MLD \times C_{SSL} + MLD \times C_{ML}}{MLD + \Delta MLD} \tag{5}$$

$\Delta MLD$ is the increase in the ML depth. $C_{ML+}$ is the tracer concentration after entrainment; and $C_{ML}$ is $C$ in the ML before entrainment. Detrainment changes the concentration of a variable C as:

$$C_{SSL+} = \frac{\Delta MLD \times C_{MLD} + SSLD \times C_{SSL}}{SSLD + \Delta MLD} \tag{6}$$

$\Delta MLD$ is the reduction in the ML depth. $C_{SSL+}$ is the variable after detrainment; $C_{SSL}$ is $C$ in the subsurface layer before detrainment.

## 2.2 Biogeochemical model

Cycling of N, Si and Fe in the ML and the SSL are modeled through a simplified food web, and through interaction between the layers and interaction with the boundary conditions (Fig. 4). The ecosystem model is also forced by changes in the ML depth and in solar irradiance, which both affect the photosynthetically active radiation (PAR). The model contains a limited number of state variables. In the ML: diatoms ($D$), nano- (and micro-) phytoplankton ($P_f$) and coccolithophores ($P_c$) represent the phytoplankton community; microzooplankton ($Z$) and mesozooplankton ($MZ$) represent the zooplankton community. In the ML and in the SSL, nitrate ($N$, $N^*$), silicate ($Si$, $Si^*$) and iron ($Fe$, $Fe^*$) are present. Two classes of detritus are defined in

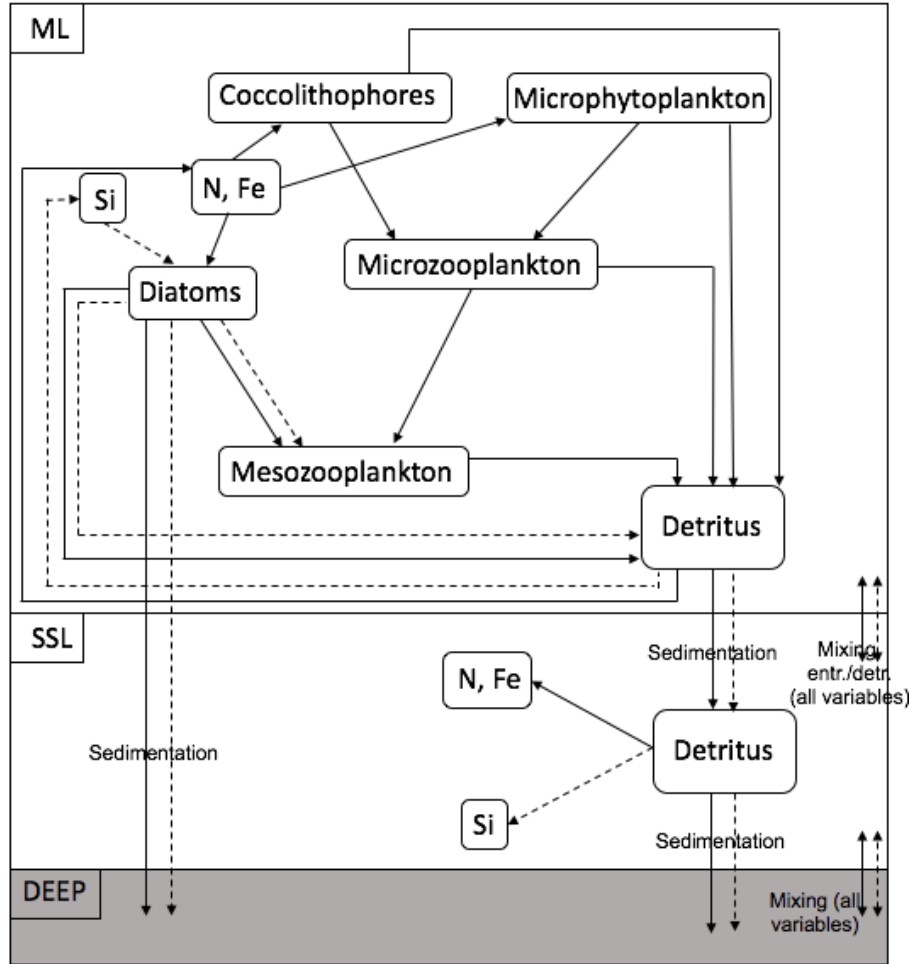

**Figure 4.** Structure of the ecosystem within one box model. The full arrows represent the flow of nitrogen and iron, the dashed arrows represent the flow of silicate

the ML and the SSL ($Dt_N$, $Dt_N^*$) and ($Dt_{Si}$, $Dt_{Si}^*$). Variables with a $^*$ are SSL variables. Nitrate has units of $mmol\,N\,m^{-3}$, silicate has units of $mmol\,Si\,m^{-3}$ and iron has units of $\mu mol\,Fe\,m^{-3}$. All other state variables except the silicate detritus pool $Dt_{Si}^*$ have units of $mmol\,N\,m^{-3}$.

### 2.2.1 Phytoplankton growth rate

5  Smaller pico- and nanophytoplankton species are dominant throughout the year in the SO (Smetacek et al., 2004). Increased biomass levels compared to this small background concentration are due to blooms of larger phytoplankton species, mainly diatoms (Smetacek et al., 2004). The biological part of the model contains only three phytoplankton species: diatoms, nano- (and micro-) phytoplankton and coccolithophores. Nanophytoplankton and coccolithophores are grazed by microzooplankton.

Microzooplankton and diatoms are a food source for mesozooplankton. Each phytoplankton species i has a realised growth rate $\mu_i$ calculated as $\mu_i = \mu_{max,i} \times \phi \times \psi$ (Tyrrell and Taylor, 1996). $\psi$ is a light limitation factor. The light limitation factor $\psi$ has a Michaelis-Menten shape and is calculated as (Tyrrell and Taylor, 1996):

$$\psi = \frac{1}{30} \sum_{i=1}^{30} \left( \frac{I_z}{I_z + I_h} \right) \tag{7}$$

The ML is subdivided into 30 equal layers. At each depth $z = (i - 0.5) \times MLD/30$ the light intensity $I_z$ is calculated. At each of the 30 layers $\psi$ is computed. An averaged value of $\psi$ over the ML is used.

About $50\%$ (the infrared component) of surface solar radiation ($I_{surf}$) never penetrates to any great depth. Properties of the remaining $50\%$ are highly dependent on the wavelength. It is common to use averaged light characteristics. However, a distinction is made between deeply penetrating green light and much less deeply penetrating red light. The light intensity at a certain depth z is calculated as (Taylor et al., 1991):

$$I_z = 0.25 I_{surf} e^{-k_r z} + 0.25 I_{surf} e^{-k_g z} \tag{8}$$

$k_r$ and $k_g$ are the extinction coefficients for red and green light respectively.

$\phi$ is a nutrient limitation factor using Michaelis-Menten calculated as:

$$\phi = min \left( \frac{N}{N + N_{h,i}}, \frac{Si}{Si + Si_{h,i}}, \frac{Fe}{Fe + Fe_{h,i}} \right) \tag{9}$$

with $N_{h,i}$ the half saturation constant for nitrate uptake, $Si_{h,i}$ the half saturation constant for silicate uptake and $Fe_{h,i}$ the half saturation constant for iron uptake (see Table 2).

### 2.2.2 Phytoplankton stoichiometry

Phytoplankton and zooplankton are modelled in units of $mmol\, N\, m^{-3}$. In order to quantify the effect of biology on other variables like Fe and Si, it is necessary to know the stoichiometry of the phytoplankton. Because of the critical role of iron, it is important to use a realistic nitrate to iron ratio (extended Redfield ratio). The Southern Ocean Iron Experiments (SOFeX) (2002-2003) (Twining et al., 2004) investigated the element stoichiometry of individual plankton cells before and after iron fertilisation. Before fertilisation the N:Fe ratio within diatoms was about 23 000:1; in flagellates it was similar. After fertilisation the ratio had decreased to about 7500:1. This suggests that phytoplankton use relatively more iron when it is widely available. Analysis of an algal culture revealed N:Fe ratio's of 4000:1 in diatoms and 2000:1 in flagellates (Ho et al., 2003). The variation is considerable and it is a crude assumption to assume a constant N:Fe ratio. Thus, the N:Fe ratio is parameterised as:

$$N:Fe = 26000 - \frac{23500 \times Fe}{1.2} \tag{10}$$

where Fe denotes the iron concentration in $\mu mol\, m^{-3}$. The N:Fe ratio as found in the model run ranges between 15800:1 and 25900:1. The ratio is rather high, but iron is considered a limiting nutrient in vast areas of the SO (Martin et al., 1990), such that these high values are justified. Equally important is the Si:N ratio. It is thought to depend on iron availability, with

higher silicification when iron is scarce (Smetacek et al., 2004). Ratios higher than 4:1 have been recorded in *Fragilariopsis Kerguelensis* (Assmy et al., 2013; Timmermans et al., 2004). As in the model study by Pasquer et al. (2015) a parameterisation for the Si:N ratio will be used:

$$Si : N = 4 - \frac{3 \times Fe}{1.2} \tag{11}$$

where Fe again indicates the iron concentration in $\mu mol\, m^{-3}$. The ratio is restricted to values between 1 and 4.

### 2.2.3 Model parameters

Model parameters are chosen within reasonable boundaries, based on literature (Oguz and Merico, 2006; Merico et al., 2006, 2004; Pondaven et al., 2000; Fasham, 1995; Pondaven et al., 1998; Taylor and Joint, 1990; Taylor et al., 1991, 1993). The phytoplankton mortality rate is constant. Grazing rates and grazing preference depend on absolute values of the prey biomass according to Pondaven et al. (2000). Large diatoms with thick frustules are known to be prone to rapid sinking, leading to opal dissolution occurring at greater depths, on average, than N and Fe remineralisation. For that reason, a greater proportion of the sinking Si is returned to solution in the deepest box of the model than is N and Fe. Attached and free coccoliths are represented in the model following Tyrrell and Taylor (1996) or Merico et al. (2006). We refer to Table 2.

## 2.3 Numerical implementation

Each station has two active layers: the ML and the SSL. A meridional section from 65°S to 40°S is defined with 30 stations, resulting in a meridional resolution of less than 1°. The model time step is $\Delta t = 10$ min. The rate of change of a certain state variable per $m^3$ is calculated using the forward Euler method. The simulation runs from 2 January 2008 to 30 December 2012. The temporal resolution of the model forcing is 3 days. The results of a 25-year simulation with cycled 5-year forcing are used as initial conditions for the final model run. For a full description of the model equations see Appendix A.

## 3 Overview of the datasets

### 3.1 Boundary conditions

The boundary conditions for the model are defined at the base of the SSL of each station along the meridional section. Boundary conditions are only required for nutrients, and they are fixed in time and space. The Global Ocean Data Analysis Project version 2 (GLODAPv2) (Olsen et al., 2016) contains data of about one million seawater samples from all over the worlds ocean, collected during almost 800 cruises between 1972 and 2013. The boundary conditions for Si and N at a specific station are obtained by averaging all available data in a zonal band from 20°E and 120°E, 50° to the east and to the west of the KERFIX longitude (Fig. 5 (a) and 5 (b)).Averaging over such a vast area is required because of the limited amount of data available. By using a large range, we assure that each latitude is sufficiently represented and that the influence of possible unrepresentative measurements (due to whatever reason) is levelled out. Furthermore, the model tries to come to conclusions

on the SO in general. From that point of view it is reasonable to use a larger area. GLODAPv2 does not contain iron data. Iron is a critical variable when investigating the SO. Therefore, it is important to use good quality data. Two datasets are combined. The Tagliabue et al. (2012) dataset contains Fe measurements across the global oceans from the years 1978 - 2009. The GEOTRACES first intermediate data product is a platform providing data from cruises around the world (Mawji, 2015). The iron bottle data collected during 7 different cruises in the SO (GIPY06, GIPY05, GIPY04, GIPY02, GI04, GA02, GPpr02) provide 3216 sample points collected between 2007 and 2011. To define the iron boundary condition, the available data are averaged over the entire zonal band (Fig. 5 (c)). The boundary conditions are specified with a limited amount of data collected all over the open SO over a large timespan. The zonal and temporal dimensions of the boundary conditions have therefore been averaged out for iron.

## 3.2 Model forcing

The Biogeochemical Southern Ocean State Estimation (B-SOSE) dataset (Verdy and Mazloff, 2017) is used for the model forcing (available at *http://sose.ucsd.edu*). B-SOSE is a model-generated best fit to SO observations - an observationally constrained solution to the MIT general circulation model. It has a horizontal resolution of 1/6°. Observations used in B-SOSE include Argo profiles, CTD measurements, mooring data, satellite measurements, etc. As it is observation-based, the results are more uncertain in regions with limited observational coverage (Mazloff and National Center for Atmospheric Research Staff, 2016). The B-SOSE simulation provides data from 2 January 2008 to 30 December 2012 with a temporal resolution of 3 days. This resolution makes this dataset suitable for model forcing. The biological part of the model is forced with a regular seasonal cycle, specifically with seasonal variations in the ML depth and in solar irradiance. A property difference-based criterion will be used to define the ML depth (de Boyer Montégut et al., 2004). Often a temperature based criterion, $\Delta T = 0.2°C$ or a density based criterion $\Delta \rho = 0.03 \, kg \, m^{-3}$ is chosen. In our model the $\Delta \rho = 0.03 \, kg \, m^{-3}$ density criterion is applied to the B-SOSE dataset. While the boundary conditions are an average in the zonal direction, the ML is defined at 30 stations between 65°S and 40°S at 67°E, a longitude which is close to the KERFIX time-series station. The daily averaged solar irradiance at the surface is calculated from the position of the sun as a function of day of year and latitude (Kirk, 1994). Cloud cover data is obtained from the NCEP/NCAR 40-year reanalysis project (Kalnay et al., 1996). Daily long-term average values over 40 years are used, and hence cloud cover is representative of an average year. As a result of this, the solar irradiance forcing in the model is the same for each year. Highest irradiance is found in January, lowest in July

## 3.3 KERFIX timeseries

The KERFIX station is located in the SO at 50°40'S, 68°25'E, south of the Polar Front and southwest of the Kerguelen Islands. Temperature, salinity and oxygen were measured between January 1990 and March 1995. Nutrients and chlorophyll data are available for the period January 1992 - December 1994. The frequency of sampling was once per month. The KERFIX program was the first 'offshore' program for regular multi-year acquisition of data in the SO (Jeandel et al., 1998). An important motivation for the KERFIX program was to increase the understanding of processes that control primary production in the area, which is rather low despite the high nutrient availability (Jeandel et al., 1998). The KERFIX site is considered a representative

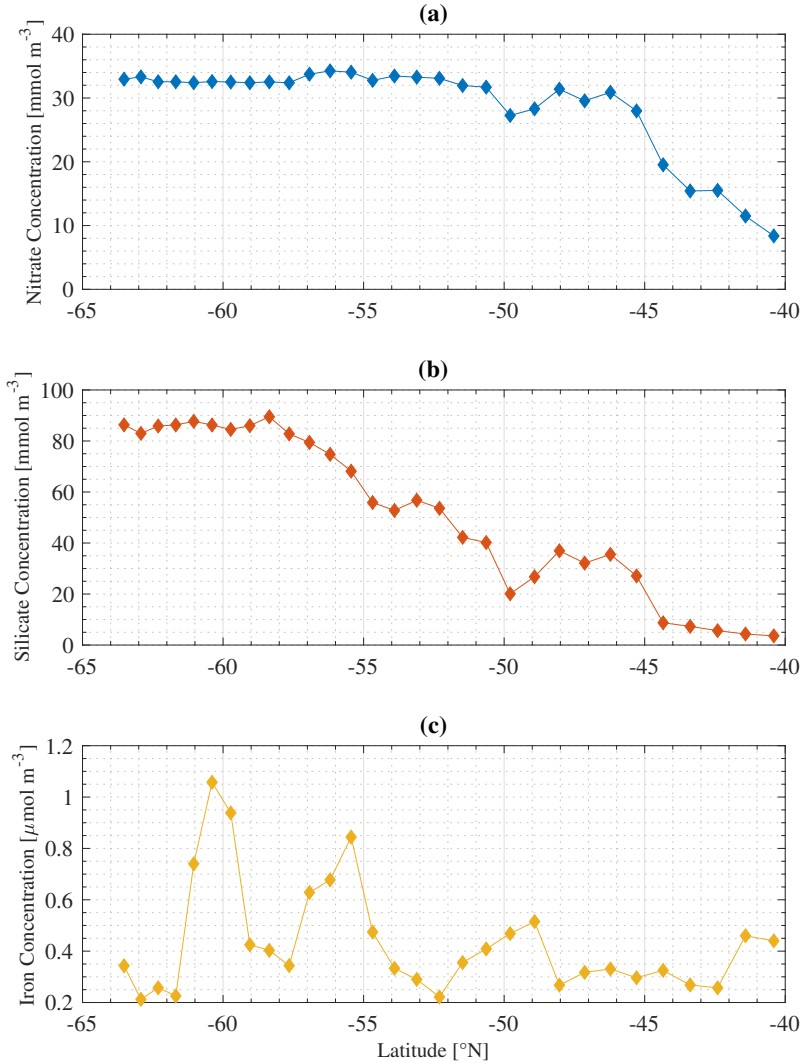

**Figure 5.** Nitrate (a), silicate (b), and iron (c) lower boundary conditions along the averaged model meridional section

location for the open SO (Louanchi et al., 2001), which is a HNLC region (Tréguer and Jacques, 1992). The KERFIX dataset is particularly useful because it enables validation of model results, forcings and boundary conditions with observational data for at least one station of the model: a reasonable comparison at one station increases the trust in the entire model. The KERFIX dataset is one of the only datasets that provides information on nutrients and phytoplankton concentrations over a

5   longer timespan. This is what makes it a very useful dataset for validating our model's performance. A perfect match between

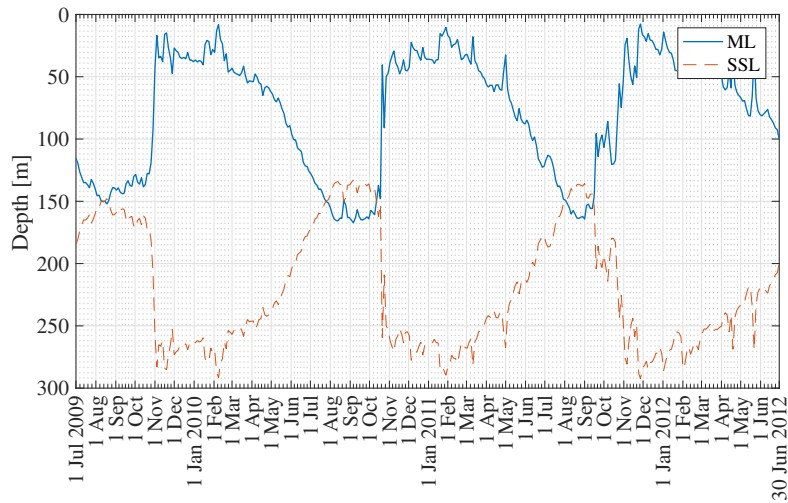

**Figure 6.** Mixed layer and Subsurface layer thicknesses at station 18 from 1 July 2009 to 30 June 2012

the KERFIX timeseries and the model is not to be expected for several reasons. First of all, the KERFIX timeseries covers a different timeframe than the model. Secondly, KERFIX is a local timeseries with a monthly sampling rate while the model uses averaged boundary conditions. The point of comparing model results with KERFIX data is therefore not to completely reproduce the KERFIX dataset, but to demonstrate to the reader that the model generates reasonable results for the purpose
intended.

The average nitrate concentration at KERFIX as measured between 1992 and 1994 at 300 m depth (the depth at which the boundary condition is defined at station 18) was 31 $mmol\,m^{-3}$. This compares well with the GLODAPv2 boundary condition for N (Fig. 5 (a)). The average silicate concentration at KERFIX was 39 $mmol\,m^{-3}$ which is again very close to the GLODAPv2 boundary condition for Si (Fig. 5 (b)).

Fig. 6 shows the ML depth from 2 January 2008 to 30 December 2012 at KERFIX based on the $\Delta\rho = 0.03\,kg\,m^{-3}$ density criterion applied to the B-SOSE dataset (see section 3.2). The seasonality is clear, with deepest mixed layers in late austral winter (150 - 200 $m$) and shallowest mixed layers in austral summer (50 $m$). A relatively rapid decrease in ML depth is visible in the second half of October (early austral spring). This is in line with observations at KERFIX between January 1990 and March 1995 (Louanchi et al., 2001). The ML depth is of similar magnitude, and a rapid decrease at the end of
October is observed as well. The high-frequency variability is attributed to events of strong re-stratification in austral winter, and de-stratification by storms in austral summer.

## 4 Results

### 4.1 Inter-annual variability at KERFIX

The permanently open ocean zone is a vast and heterogeneous area in the SO. It is a remote and rough region, and it is thus relatively poorly sampled (Smith et al., 2000). Due to limited data availability, there is no added value in expanding the model with a zonal component; thus, the model represents a zonal average of the SO. Despite the attractive simplicity of this assumption, it makes comparing model results with one localised sampling dataset (obtained during a specific cruise, or satellite mission, acquired at a certain time in year, or using specific methods, etc.) difficult. While the model results can be a representative average, localised observational data are not necessarily so. With this caveat, model results at station 18 (the KERFIX latitude) are compared with the KERFIX observational data between January 1990 and March 1995, with satellite data and with other model results.

Fig. 7 (a) and (b) show the modelled phytoplankton and zooplankton biomass between 1 July 2009 and 30 June 2012 at station 18. Diatoms are the first phytoplankton species to bloom. The peak of 1.4 - 1.6 $mmol\,N\,m^{-3}$ occurs mid-November. A second diatom peak occurs around mid-March, and is significantly smaller than the main diatom bloom, with a concentration up to 0.5 $mmol\,N\,m^{-3}$. Nanophytoplankton biomass reaches a maximum concentration of 0.9 - 1 $mmol\,N\,m^{-3}$ at the beginning of January. Coccolithophores never attain high concentrations. However, just like diatoms and nanophytoplankton they are able to sustain biomass from the beginning of December to the end of May.

The zooplankton biomass closely follows the phytoplankton curves. Mesozooplankton concentration exhibits a bloom shortly after the diatom bloom. Following this bloom mesozooplankton feed on microzooplankton. This factor, together with lower nanophytoplankton and coccolithophore biomass keeps the microzooplankton biomass rather low.

The biomass of phytoplankton is expressed in $mg\,chl\,a\,m^{-3}$. A Redfield C:N-ratio of 6.625 $mol\,mol^{-1}$ is used (Redfield, 1934). Gall et al. (2001) investigated the response of the phytoplankton community to in-situ iron fertilisation in the SO waters (SOIREE). One of their findings was that iron availability influenced the carbon to chlorophyll-a ratios. Prior to fertilisation the algal community had a mean carbon to chlorophyll-a ratio $< 100$ $g\,g^{-1}$(Diatoms: 94 $g\,g^{-1}$, nanophytoplankton: 40 $g\,g^{-1}$). After fertilisation algal community mean carbon to chlorophyll-a ratio had decreased to 40 $g\,g^{-1}$. Similarly, temperature, light and nutrient availability affect the carbon to chlorophyll-a ratio (Cloern et al., 1995), which can vary from 100 $g\,g^{-1}$ to 30 $g\,g^{-1}$. Pondaven et al. (2000) used a value of 60 $g\,g^{-1}$ for nanophytoplankton, and 45 $g\,g^{-1}$ for microphytoplankton, in their model study at KERFIX in the Southern Ocean. In our model a carbon to chlorophyll-a ratio of 100 $g\,g^{-1}$ will be used for each phytoplankton species, independent of temperature, light or nutrient concentrations. Note that the chlorophyll-a concentration is inversely proportional with the ratio. So using a ratio of 80 $g\,g^{-1}$ instead of 100 $g\,g^{-1}$ will increase the chlorophyll-a concentration with 25%.

Modelled total chlorophyll-a concentration in the ML between 1 July 2009 and 30 June 2012 is presented in Fig. 7 (c) (solid line), with the dashed line showing assimilated daily chlorophyll-a data from the NASA Ocean Biogeochemical Model (NOBM) (Goddard Modeling and Assimilation Office (GMAO), 2014). Monthly satellite data for chlorophyll-a are marked by triangles (MODISA), and the comparison with in-situ KERFIX data is shown with diamonds. There are some similarities

and some discrepancies. Whereas the model finds no biomass in austral winter, the satellite data and NOB model find biomass throughout the year. No blooms are observed in the satellite and NOBM curves. The satellite data is monthly averaged. This can lead to averaging out of possible short blooms. Apart from the difference in austral winter and the lack of a bloom in the NOB model and satellite data, the biomass concentrations in austral summer and autumn are similar. Our model attains reasonable results compared with the in-situ KERFIX data: a phytoplankton peak around mid-November, and low but persistent chlorophyll-a values in summer and autumn. However, as for the satellite data, there is an observed winter concentration of phytoplankton that is not captured by our model. This discrepancy was not rectified because most nutrient removal occurs in the spring and summer.

Pondaven et al. (2000) used a similar 1-D model to calculate the phytoplankton biomass in the ML at KERFIX, using physical forcing between 1 July 1990 and 30 June 1991. They found a bloom of 1.5 $mg\,chl\,a\,m^{-3}$ on 15 November. Their results were compared to actual chlorophyll-a data obtained in 1992, 1993 and 1994 in which peaks of about 0.8 $mg\,chl\,a\,m^{-3}$ were observed. Similarly, Pasquer et al. (2015) used the SWAMCO model at the KERFIX location to find a bloom of about 1.5 $mg\,chl\,a\,m^{-3}$ in spring. However, the bloom lasted significantly longer than found in our model or in the model by Pondaven et al. (2000).

In situ measurements at KERFIX (1991-1995) revealed the occurrence of blooms at that location, despite the absence of blooms in satellite data. Highest recorded in situ values were about 0.8 $mg\,chl\,a\,m^{-3}$. It is possible that due to the monthly frequency of sampling, higher values were missed out. Our modelled peak of 1.5 - 1.6 $mg\,chl\,a\,m^{-3}$ is thus realistic. The only characteristic unique to our model is the zero biomass in austral winter which, as mentioned, was not rectified.

Fig. 8 (a), (b) and (c) present the modelled nutrient concentrations in the ML. It is important to recall that the period 2009-2012 (in our model) is compared with the period 1993-1995 (in KERFIX). Diamonds denote monthly averaged observations at KERFIX between July 1993 and July 1995. Nutrient levels are high in winter due to entrainment, diffusion and upwelling/advection (the three processes responsible for re-setting winter concentration in the model, although as shown below entrainment is the most important). As soon as diatoms start blooming, nutrients are drawn down rapidly and reach their lowest values during spring/summer, after the nanophytoplankton bloom. The modelled October nitrate concentration is 26 $mmol\,m^{-3}$. Biological utilisation reduces this concentration to 23 $mmol\,m^{-3}$ in May, a decrease of 3 $mmol\,m^{-3}$. During in situ measurements at KERFIX, peak winter concentrations of about 28 $mmol\,m^{-3}$ and summer minima of about 24 $mmol\,m^{-3}$ were observed (Louanchi et al., 2001). The difference between summer and winter is relatively similar between our model results and the in-situ KERFIX data, although of slightly larger amplitude in the data. The overall modelled nitrate cycle compares well with the KERFIX observations too. The modelled October silicate concentration is 20 $mmol\,m^{-3}$. Biological utilisation reduces this concentration to 9 $mmol\,m^{-3}$ by December, a decrease of 11 $mmol\,m^{-3}$. In-situ KERFIX data (Louanchi et al., 2001) showed a winter concentration of about 20 $mmol\,m^{-3}$ and a summer concentration that is about 12 $mmol\,m^{-3}$ lower. Despite the reasonable agreement between these values, there is a clear temporal offset between the model's seasonality and the observed seasonality at KERFIX for Si concentrations. The timing of highest concentration is well modelled, as is the rapid depletion of Si at the start of spring as well. The restoring of Si after the spring bloom is rather different. Observations show absolute minima at a time in which modelled concentrations are recovering. It appears that the

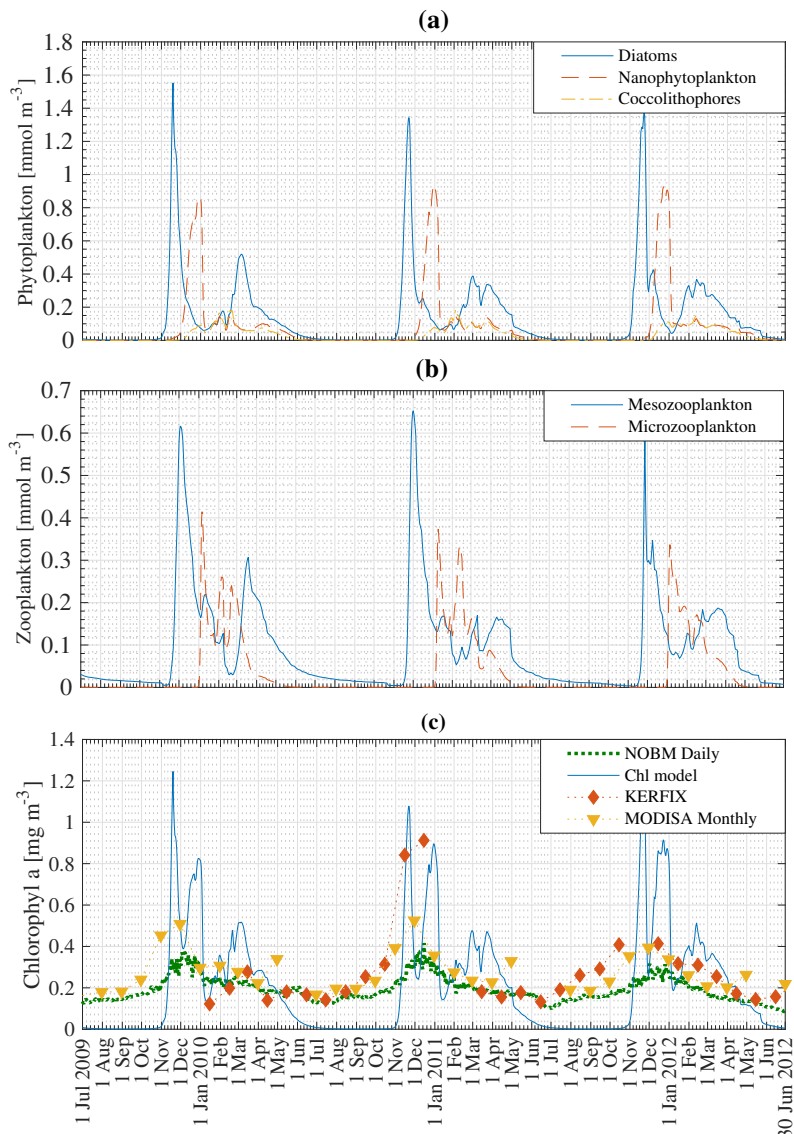

**Figure 7.** Model result: diatom, nanophytoplankton and coccolithophore concentration (a), mesozooplankton and microzooplankton concentration (b), and chlorophyll-a concentration compared to KERFIX in-situ data (♦), to MODISA satellite data (▼) and to NOBM results (··) (c) in the mixed layer at station 18 from 1 July 2009 to 30 June 2012. KERFIX in-situ data from January 1992 to June 1994

model misses the actual Si minimum. This likely means that the diatom bloom is cut off too soon, and that a longer-lasting bloom as found in the model study by Pasquer et al. (2015) is more realistic. Modelled concentrations of the micronutrient iron are low throughout the year. Winter concentrations reach $0.12\ \mu mol\,m^{-3}$. Primary production reduces iron levels to almost

zero in summer. Consequently phytoplankton cannot sustain biomass and the modelled concentration of phytoplankton reaches zero. As discussed earlier, this is not in line with observations and other model studies. The reason for this discrepancy might be that remineralisation in the model is underestimated. Tagliabue et al. (2014) suggest that iron concentration is replenished by a boost of entrainment in winter leading to phytoplankton blooms and that a limited amount of iron is sustained throughout the rest of the year by efficient remineralisation, allowing biomass to be sustained. Another explanation for the lack of biomass in winter is the low light availability during large parts of the year. As the model uses an average PAR for the entire ML, it is likely that PAR at the surface is underestimated by the model.

The nanophytoplankton bloom is evidence that there are still sufficient nutrients available in the ML after the diatom bloom. This raises the question of what it is that limits further diatom production. Diatom growth can be limited by insufficient light, significant grazing pressure, Si limitation, or Fe limitation in our model. At that time of the year, it is unlikely that diatom growth is limited by lack of light. According to our model, diatoms are not limited by Si either, as concentrations are sufficiently high throughout the year (Fig. 8 (b)), in line with KERFIX data. This is in contrast with results of the model study by Pondaven et al. (2000), who found that diatoms are prevented from blooming at the surface due to a lack of Si. Their 1-D model (Pondaven et al., 2000) has a high vertical resolution, and is able to capture the vertical structure of nutrient distributions. This explains why their study obtains a silicate concentration as low as $3\ mmol\ m^{-3}$ at the surface. Note also that they use a relatively high half saturation constant for Si uptake, $Si_h$, $8\ mmol\ m^{-3}$, which is close to the range of measured values (12-27 $mmol\ m^{-3}$ have been reported in the open SO (Caubert, 1998; Jacques, 1983)). While in our model a substantially lower $Si_h$ of $1\ mmol\ m^{-3}$ is adopted, our results are not sensitive to this choice. Indeed, running the model with $Si_h = 8\ mmol\ m^{-3}$ does not change the diatom bloom concentration significantly, indicating that in an averaged ML diatom growth is not limited by insufficient Si. Pondaven et al. (2000) suggest that the high values of the half saturation constant for Si uptake $Si_h$ are actually a consequence of iron limitation, such that their model (which does not explicitly represent Fe) accounts for iron limitation through a high value of $Si_h$. Si limitation then becomes a consequence of iron limitation. In our model, diatom growth is limited by iron. Nanophytoplankton are capable of assimilating iron more efficiently. However, the mesozooplankton bloom follows the diatom bloom closely, suggesting that intensive grazing limits diatom growth. To substantiate this interpretation, we conduct a model run with reduced grazing pressure in the model (no mesozooplankton); this reveals that diatom growth is indeed limited by a lack of iron. When the iron concentration decreases below $0.05\ \mu mol\ m^{-3}$, the diatom abundance is reduced. The diatom population decreases more rapidly when mesozooplankton are present in the model, but the presence of mesozooplankton does not significantly alter the distribution of phytoplankton over the season. When diatoms disappear, nanophytoplankton use the remaining of the iron until it is depleted to an even lower level where it becomes limiting to them as well. When iron concentrations increase around February - March, diatoms are able to bloom for a second time, albeit in a less pronounced manner.

## 4.2 Nutrient distribution along an averaged section in the SO

The macronutrient gradient in the upper limb of the SO is reproduced in the model (Fig. 9 (a) and (b)). To generate these figures, winter is defined here as the period (1 July 2010 and 30 June 2011) in which nutrient concentrations in the water are

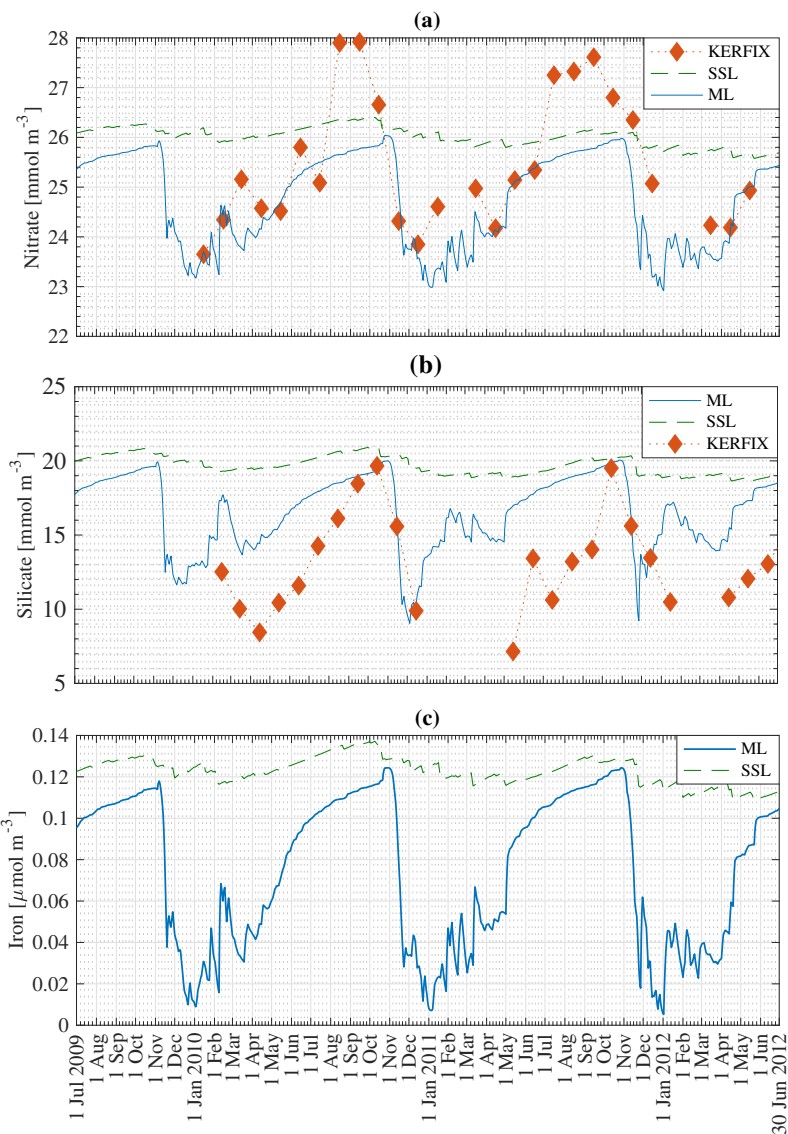

**Figure 8.** Model result: nitrate (a), silicate (b), and iron (c) concentration in the mixed layer (–) and in the subsurface layer (- -) compared to KERFIX in-situ data (♦) at station 18 from 1 July 2009 to 30 June 2012

highest, whereas summer is defined as the period (2 January 2010 and 30 December 2010) in which nutrient concentrations are lowest. The winter nitrate concentration at the Southern Boundary ($\sim 60°$S) is about 30 $mmol\,m^{-3}$, decreasing approximately linearly to 24 $mmol\,m^{-3}$ at 45°S, i.e. a reduction of 6 $mmol\,m^{-3}$ over 15 degrees of latitude (0.4 $mmol\,m^{-3}$ per degree latitude). From 45 to 40°S the gradient is stronger, with a reduction of 10 $mmol\,m^{-3}$ over 5 degrees of latitude (2 $mmol\,m^{-3}$

per degree latitude). The winter nitrate gradient is overall 0.7 $mmol\,m^{-3}$ per degree latitude. The gradient is a lot stronger for silicate (Fig. 9 (b)) . At the Southern Boundary, 70 $mmol\,m^{-3}$ is reached in winter. Concentrations (also in winter) reduce to zero at 40°S, i.e. a gradient of about 3 $mmol\,m^{-3}$ per degree latitude. These results are in line with observations made along sections in the SO (Section 1 Introduction and Tréguer and Jacques (1992); Pollard et al. (2002); Assmy et al. (2013); Smith et al. (2000); Read et al. (2002)).

Note that the deep nitrate concentration north of 44°S is lower than the modelled winter/summer nitrate concentrations. Because there are no observational data that support this claim, it seems that the modelled nitrate concentration north of 44°S is overestimated or that the deep water concentration is underestimated. This should be taken into account when assessing the results north of 44°S further in this work.

There is no substantial gradient in the iron concentration, irrespective of season (Fig. 9 (c)). The summer gradients for N and Si are similar in shape to the winter gradients. The difference between the summer and winter gradient provides a good indication of primary productivity at a certain station. For iron, the summer concentration is near-zero all along the section, indicating the limiting nature of the micronutrient in the model. The winter concentration of iron is correlated with the summer concentration of both N and Si. If abundant iron is present, more Si and N can be assimilated, leading to larger differences between summer and winter concentrations of these macronutrients. The dotted line in each of Fig.9 (a), (b) and (c) shows the boundary condition as used in our model. This indicates the existence of a connection between the boundary condition concentration and the ML concentration. In summary: the winter concentration (gradient) of N and Si in the ML depends strongly on the deep water concentration (gradient) of N and Si. The summer concentration of N and Si also depends strongly on the winter concentration of iron, and on the amount of biological production that it enables during the growing season. The model thus suggests that a crucial factor in understanding the observed gradients in surface nutrient concentration is the concentration at depth along a section.

### 4.3 Processes restoring winter concentrations of N and Si in the mixed layer

Nutrient concentrations in the ML are depleted by biology (use of nutrients by phytoplankton and sinking of diatoms and detritus) and restored due to four processes in our model: entrainment, diffusive mixing, remineralisation and advection/upwelling. The absolute contribution of each process to the restoration of nitrate in the ML for station 18 is shown in Fig. 10 (a). Fig. 10 (b) shows the absolute contribution per season. It is the integration of Fig. 10 (a) over time for Spring 2010, Summer 2010-2011, Autumn 2011 and Winter 2011. Number in brackets are the net increase of nitrate in the mixed layer in $mmol\,m^{-3}$. Biology is the major sink of nitrate during spring and summer. Entrainment is the major source of nitrate, followed by remineralisation as a second important source of nitrate. The absolute difference between the maximum and minimum N concentration at station 18 over that time period is about 3 $mmol\,m^{-3}$ (Fig. 8 (a)) while the sum of all processes restoring the nitrate concentration adds up to about 16 $mmol\,m^{-3}$. It is clear that even during periods of heavy primary production, nitrate is constantly replenished (partly because physical processes only have an effect on the surface concentration when the introduced water has a different concentration). It is during late summer and especially during autumn that we have a net increase of nitrate in the MLD. The final winter concentration is then reached during winter.

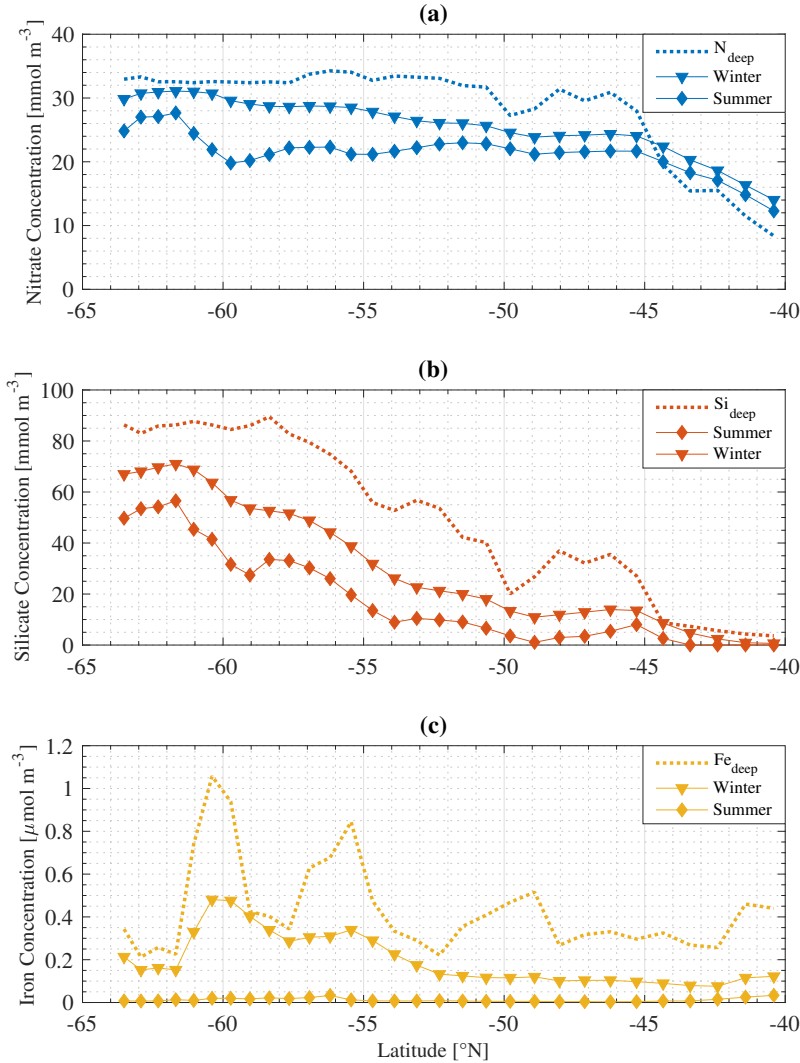

**Figure 9.** Model result: summer (♦) and winter (▼) nitrate (a), silicate (b), and iron (c) concentration in the mixed layer along the average meridional section compared to the deep water concentration (··)

In order to quantify the importance of each process at each station over the meridional section, the relative contribution of each restoring process is calculated. The relative contribution of a process is defined here as the increase of the nutrient N, Si or Fe due to that process between 1 July 2010 and 30 June 2011, divided by the total increase of that nutrient in that period. The first of July is chosen because physical effects are small at that time, as seen in Fig. 10.

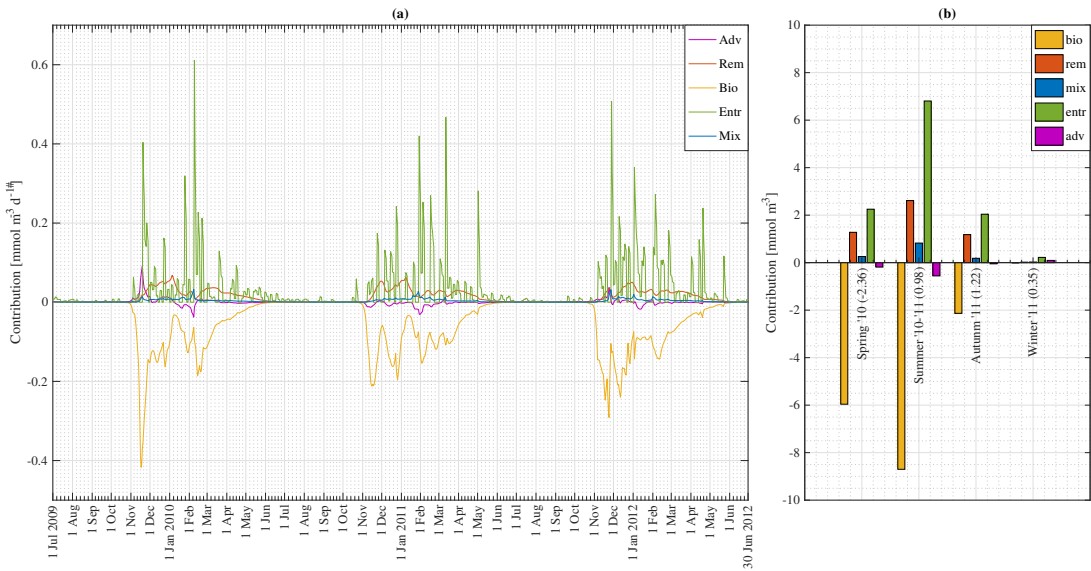

**Figure 10.** Model result: (a) absolute contribution of advection/upwelling, remineralisation, biology, diffusive mixing and entrainment to the nitrate concentration in the mixed layer at station 18 from 1 July 2009 to 30 June 2012 (b) Integration of figure (a) over the seasons: spring 2010, summer 2010-'11, autumn 2011 and winter 2011

The relative contribution of each process is quantified and visualised in Fig. 11 (a) and (b) for N and Si, respectively. Entrainment of water from the SSL to the ML is the dominant process restoring winter N concentrations in the ML: typically about 50 percent, with a peak of up to 80 percent around 48°S. Remineralisation is important as well, providing 20 to 40 percent of the nitrate depending on latitude. Diffusive mixing is of minor importance (about 10 percent). Note that this result
justifies the omission of horizontal diffusion in the model. Advection/upwelling is unimportant too. At some latitudes, advection contributes negatively; the concentration of N at 49°S is lower than the concentration at 48°S and for that reason the effect of water flowing northward from 49°S to 48°S is to lower the concentration at 48°S. From about 44°S, advection plays a more important role than entrainment. As explained earlier (Fig. 9 (a)) the concentration in the deep water is lower than that in the ML and SSL. There is a loss of nutrients from the SSL to the deep, and therefore the concentration of the water entrained
into the ML is substantially lower and entrainment is reduced. Although advection plays a more important role in that zone, it is noted that the difference between summer and winter concentrations are very small overall, indicating a low primary production compared to stations further south

There are two main findings. First, and most importantly, entrainment is the dominant process restoring N in the ML followed by remineralisation. Diffusive mixing and advection play a minor role. Second, when the concentration in the deep water is
15 low, advection is the dominant process restoring winter concentrations. However, the change in concentration from summer to winter is very limited in this area due to low primary production.

Entrainment also dominates wintertime recovery of ML silicate levels (Fig. 11 (b)). Diffusive mixing is rather unimportant, as is remineralisation. As there are only two significant processes, it is found that the percentage contribution of advection/upwelling is nearly a replica of the entrainment term.

An interesting difference is noticed between N and Si. Remineralisation is not important for Si, yet it is important for N. There are two reasons for this. First, the choice of the model remineralisation parameter is different for Si and N, being twice as high for N as for Si (0.16 $d^{-1}$ and 0.08 $d^{-1}$, respectively). Second, Si sinks out of the detritus pool much faster than N does, such that less Si gets remineralised from detritus back into the ML. In the model the sinking velocity of N and Fe in detritus is equal to 5 m $d^{-1}$, while it is chosen as 50 m $d^{-1}$ for Si. These choices are incorporated in our model to represent uncoupling of N and Si export (Tréguer and Jacques, 1992; Pollard et al., 2002; Assmy et al., 2013). Particulate Si (diatom frustules) sinks faster and dissolves at greater depth, or does not dissolve and ends up in the siliceous ooze (Assmy et al., 2013).

A negative contribution of advection is noticed, especially more to the north. From a mathematical point of view, this means that the yearly average concentration in station i-1 is lower than in station i+1. The net flow of water into that station is that of water with a concentration lower than that in station i. Referring to Fig. 8, the pattern behind the advection term is reflected in the summer gradients of N and Si and hence in the iron winter concentration. At stations with high winter iron concentration, we notice a high primary production in the subsequent months; this drives an important drawdown of nutrients, lower summer nutrient concentrations and hence a lower average nutrient concentration over an entire year than in adjacent stations with lower primary production. These stations can then provide a sink for nutrients to the adjacent stations with higher nutrient concentrations.

Fig. 11 (c) shows the relative contribution of each process to the winter recovery of iron at each station. As for N and Si, entrainment is the most important process restoring the winter concentration. It is about ten times more important than diffusive mixing. This result is in line with a data study by Tagliabue et al. (2014). We see that remineralisation plays an important role as well, despite the fact that it is insufficient to sustain phytoplankton biomass throughout the year.

Fig. 12 is an extensive comparison between model results at station 29 (41°S) and at station 2 (63°S), stations with low and high Si concentrations respectively. The concentration of Si at station 29 in the deep layer, the SSL and the ML are of a different magnitude of order to those at station 2 (see Fig. 11 (b)). By comparing results, we try to gain insight in the processes responsible for that difference. (a) shows the difference between the concentration in the SSL and in the MLD. In station 2 the difference is significant with differences up to 15 $mmol/m^3$. The effect of mixing is linearly dependent on this difference (see Fig. 12 (b)). That is why mixing is a lot more important at station 2 than it is at station 29 (although unimportant in general at both stations). Fig. 12 (c) shows the chlorophyll-a production at both stations. The seasonal chlorophyll-a maxima at station 2 are significantly higher than at station 29. It is no surprise, therefore, to see that biology is a more important loss at station 2 than it is at station 29 (Fig. 12 (d)). Entrainment depends on the change in ML over time and on the difference in concentration between the mixed layer and the subsurface layer. The mixed layer depth at station 2 and station 29 are rather similar. The contributions of entrainment are not similar. This must be due to the difference in ΔSi. The difference in Si concentration

between ML and SSL is significant at station 2 and rather small at station 29. Because the mixed layer is rather similar, it is really the difference in concentration between ML and SSL that pushes entrainment to high levels at station 2. Fig. 12 (h) shows the contribution of advection and upwelling at both stations. The total contribution of advection/upwelling seems to be more or less similar for station 2 and station 29.

Fig. 12 (g) shows the cumulative contribution of each process to the Si concentration in the ML at station 2 and station 29 between 1 June 2010 and 1 June 2011. It shows that there is a lot more cycling of Si at station 2. biology is a major sink and it is mainly entrainment that replenishes Si. Primary production is low at the northern edge where Si is limiting. It is noticed that advection is equally important to restore Si concentrations as entrainment to restore Si concentrations at station 29.

Fig. 12 indicates that it is the high concentrations in the SSL and at the deep boundary, and hence the significant difference

between ML and SSL after the primary production season, that supports the high concentrations at high latitudes. The opposite is true for the lower latitudes. At lower latitudes, there is no replenishment from the deep, not due to a lack of winter mixing events (changes in MLD), but due to a lack of nutrients at depth.

## 5   Discussion

### 5.1   Role of physics in shaping meridional gradients of N and Si in the mixed layer

The observed winter gradient of N and Si along a south-to-north section is a smooth mirror image of the gradient observed in the boundary conditions for N and Si. This suggests that, if no boundary condition gradient existed, no gradient would be observed in the ML. Indeed, if the model is run with a fixed boundary condition for N ($30\ mmol\,m^{-3}$) and Si ($60\ mmol\,m^{-3}$) along the section, the winter gradient disappears, and winter concentrations of both N and Si are relatively similar along the section with a change of max $10\ mmol\,m^{-3}$ for Si and $5\ mmol\,m^{-3}$ for N along the meridional section (Fig. 13 (a)).

This strongly suggests that the observed winter meridional nutrient gradient in the ML results from the deep-water nutrient distribution. Having a nutrient gradient below the SSL is a necessary requirement for a nutrient gradient to occur close to the surface. This is especially true in winter, when mixed layers are deepest (**?**)and when mode waters form (Cerovecki et al., 2013).

Results when biology is excluded from the model suggest that biological processes are less important to reproduce a near-

surface macronutrient gradient. Fig. 13 (b) compares the standard model results with a model run where biology is omitted. Horizontal nutrient gradients persist. The overall concentration of nutrient increases, which is expected because the major sink is taken out of the model.

Concentrations do not reach deep water values when biology is excluded from the model. Why is this not so with entrainment/detrainment isolated as a prominent process. Entrainment/Detrainment is important, but it occurs only from ML to SSL

and from SSL to ML. So the total amount of nutrients is divided between SSL and ML (variable with changing ML and SSL depth). The interaction with other boxes is via advection; the interaction with the deep (boundary condition) is via diffusive mixing. Diffusive mixing plays a role, but only on larger timescales, compared to advection and entrainment ($k_{mix}$ rather

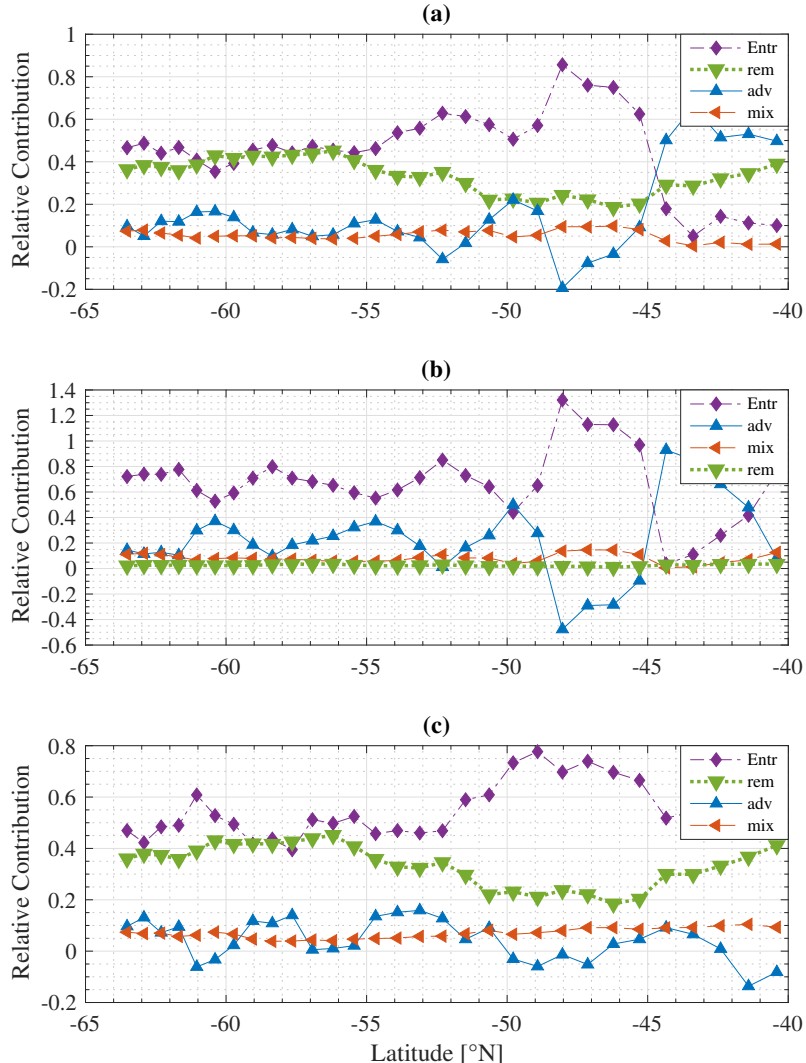

**Figure 11.** Model result: relative contribution of entrainment (♦), diffusive mixing (◄), remineralisation (▼) and advection/upwelling (▲) to restoration of the winter nitrate (a), silicate (b), and iron (c) concentration at each station along the average meridional section

low). So, ML concentration and SSL concentration are very similar without biology, but to equilibrate with the deep water concentration, the model would have to run for a longer time period.

Fig. 13 (a) and (b) indicate that there would be no ML nutrient gradient at all without a gradient at depth, and without the connection between the deep and surface waters. Fig. 11 (a) and (b) reveal that physical processes prevail in providing this

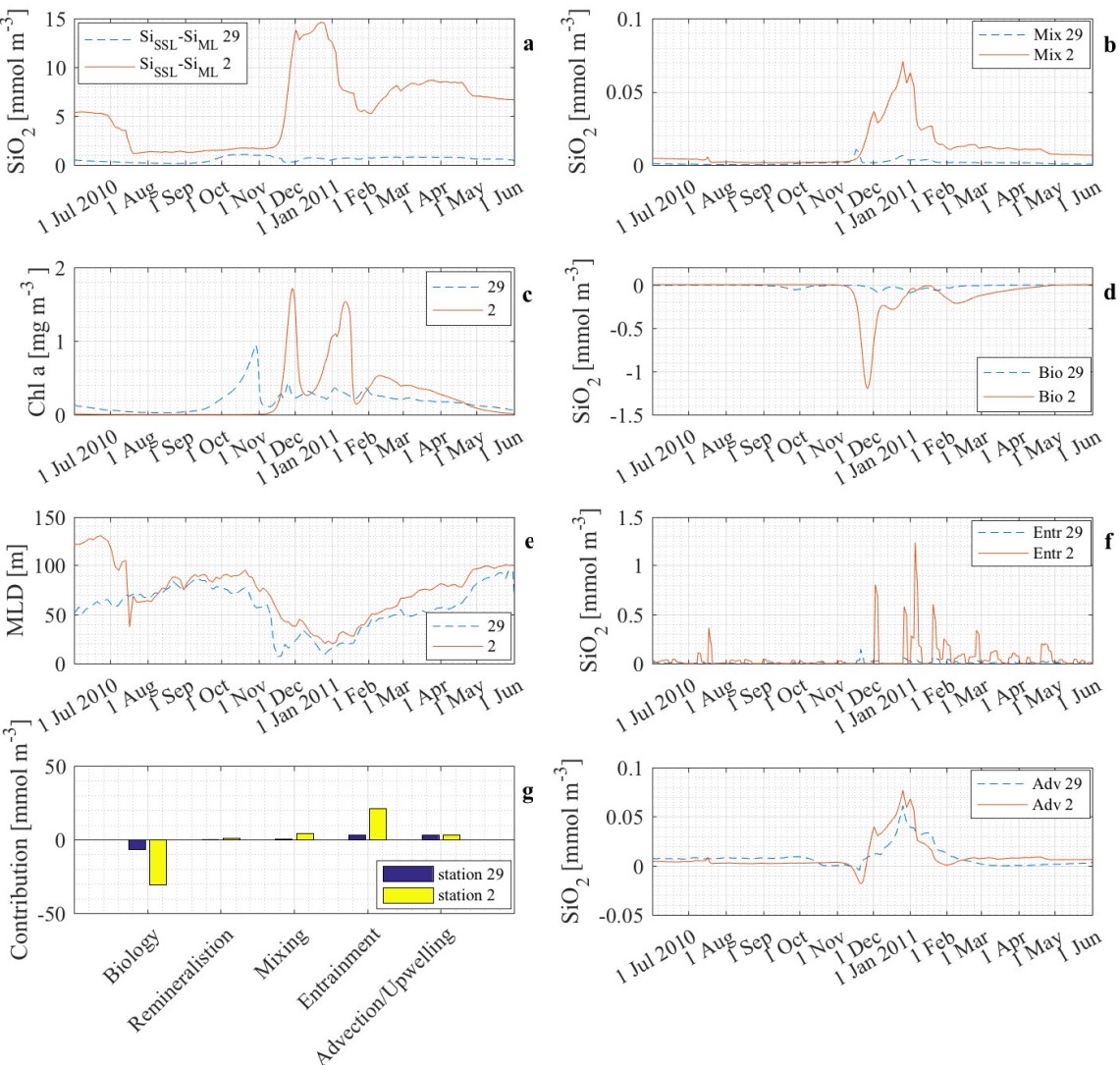

**Figure 12.** Model results at station 2 (63°) and station 29 (41°S) between 1 June 2010 and 1 June 201 - comparison: (a) $\Delta Si_{SSL-ML}$; (b) contribution of mixing to the silicate concentration in the mixed layer; (c) chlorophyll-a concentration in the mixed layer; (d) contribution of biology and remineralisation to the silicate concentration in the mixed layer; (e) mixed layer depth; (f) contribution of entrainment to the silicate concentration in the mixed layer; (g) cumulative contribution of each process from 1 June 2010 to 1 June 2011; (h) contribution of advection/upwelling to the silicate concentration in the mixed layer

connection. Thus, our model strongly suggests that entrainment (deep winter mixing) is the main factor controlling the winter concentration of Si and N in the ML. Note that we cannot explicitly demonstrate this by running the model without physics,

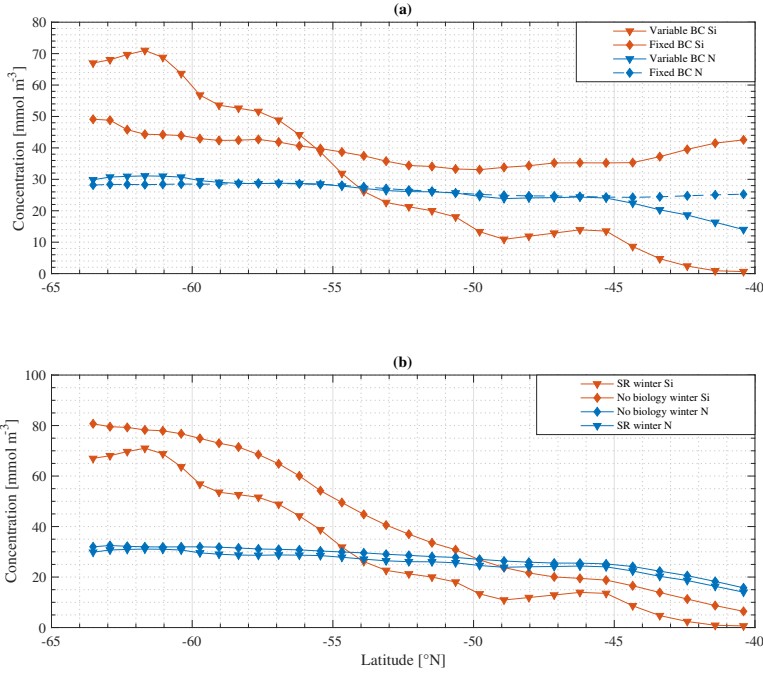

**Figure 13.** Model result: winter nitrate and silicate concentration in the mixed layer along the average meridional section - a comparison of the standard model run (SR) (▼) with a model run with constant boundary condition (♦) (a) and a model run without biology (♦) (b)

because iron would not be replenished in winter, phytoplankton and zooplankton would vanish, and N and Si would remain constant, making model results uninformative.

## 5.2 Different drivers over different timescales

Model results are found to be rather sensitive to changes in $C_{deep}$, the deep-water nutrient concentration. It is not possible to reproduce the strong Si gradient and weaker N gradient if $C_{deep}$ along the section is assumed constant, at least over the timescales of relevance to the model (i.e. timescales of years to decades over which the deep-water nutrient distribution may be assumed to be constant in time). $C_{deep}$ is, by definition, the concentration at the base of the SSL. If zonal variation in $C_{deep}$ is a necessary requirement for sustaining the nutrient concentration gradient in the ML, and if the ML gradient reflects a deep-water gradient, it is necessary to think about the processes responsible for the deep-water gradient. Our model can address the question of proximate causes of silicate depletion at the northern end of the overturning's upper limb (causes of surface silicate patterns over short timescales of years). However, our model provides no insights into the controls over the deep-water gradient on longer timescales.

We acknowledge that biological processes would eventually impact the deep ocean boundary condition in the real ocean, both because organic matter remineralisation is important in maintaining deep concentrations, but also due to mixing. The weakened surface nutrient gradient in the absence of biological uptake would soon start to impact the subsurface layer (through detrainment) and from there the deep layer due to diffusive mixing. Vertical gradients would therefore start to weaken in the absence of biology. However, we emphasize that the argument we make is about the effect of biology on horizontal gradients, not on vertical gradients. Our argument is not affected by the weakening of vertical gradients without biology. If, over time, vertical gradients were to be completely eliminated following removal of biology then surface mixed layer and deep nutrient concentrations would tend to become identical. The horizontal gradient in the surface nutrients would then become identical to the horizontal gradient in the deep nutrients. Therefore, because there is a strong north-south gradient at depth, there would continue to be one also at the surface. It can be seen that an elimination of vertical gradients does not in any way imply an elimination of horizontal gradients. It does not seem likely that removing biology can make large differences to deep nutrient concentrations over a few years. This is because the annual remineralisation fluxes at depths of several hundred metres are very small compared to the ambient nutrient concentrations below the SSL, which would make it impossible to alter deep nutrient concentration over one or a few years.

## 5.3 Robustness of the main result

Some additional model runs were carried out in order to investigate the sensitivity of the main result to some assumptions. Firstly, upwelling velocities were increased by 50% in one model run and decreased by 50% in another. The model does not explicitly model either eddies or Ekman transport and therefore makes no distinction between the different sources of the northward transport. The northward transport is calculated from conservation of mass and so depends on upwelling velocities. Therefore, the northward transport was also altered in these runs. It should be kept in mind that these alterations are most likely contradicted by some of the data to constrain the behaviour of the SOSE model.

Secondly, in reality the northwards transport is not completely restricted to the ML but rather takes place partly in the SSL as well. For one model run 80% of the total northward transport was made to occur in the ML and 20% in the SSL.

As shown in Fig. 14, applying these changes one by one to the model does not greatly affect the final result in terms of the primacy of physical processes (entrainment) over biological processes in driving nutrient patterns. For each altered model it remains true that the silicate gradient (the south-to-north gradient in the ML concentration) is more strongly affected by making the bottom boundary condition constant than it is by removing biology from the model.

## 5.4 Model performance

The model has proven to be a useful tool for better understanding the processes responsible for the nutrient distribution along an averaged meridional section in the SO. Nutrient cycles are reproduced with reasonable skill. There is room for improvement when it comes to the Si cycle, but it is not within the scope of this work to produce a completely realistic simulation, and our important results are not affected by this limitation. Phytoplankton cycles are modelled quite well. In reality, blooms might be more spread over the season than in our model, which generates a short and intensive bloom. In reality, non-negligible

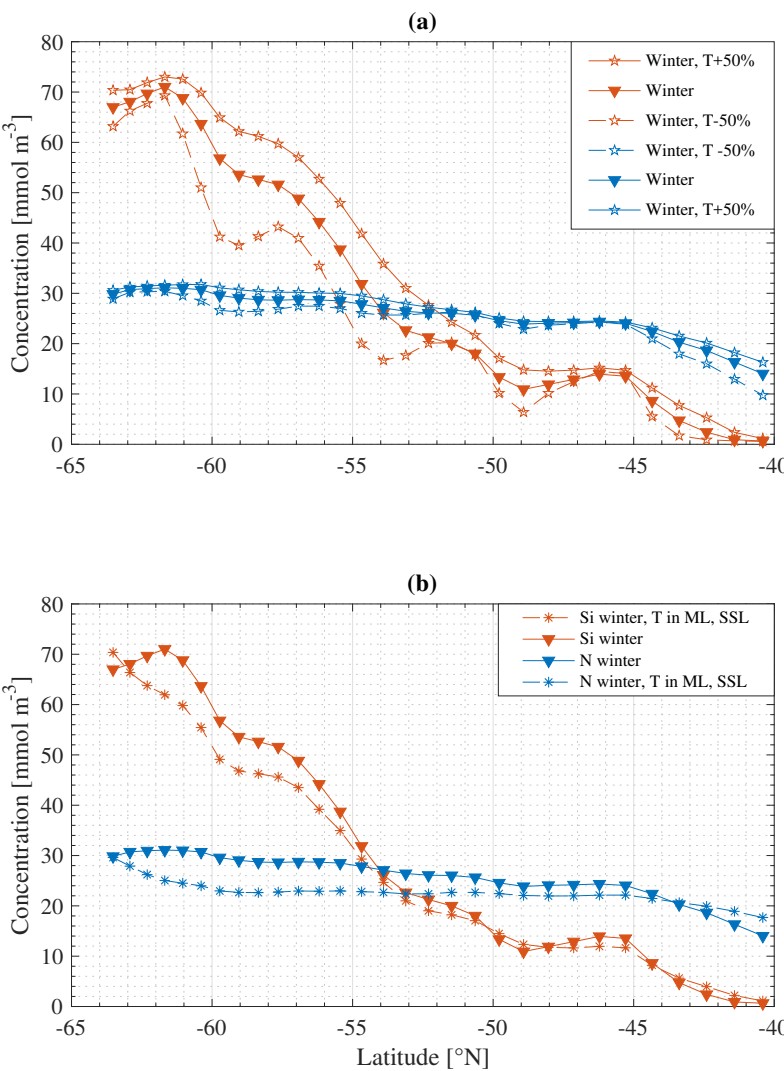

**Figure 14.** Model result: winter nitrate and silicate concentration in the mixed layer along the average meridional section - a comparison of the standard model run (▼) with a model run where transport is increased and decreased with 50% (∗) (a), and with a model run where transport is divided over the ML and SSL (∗) (b)

populations of phytoplankton appear able to survive throughout the SO winter, whereas they die away in the model. However, this discrepancy does not affect our results, given that the large majority of nutrient removal takes place in the spring and summer

A key result of our model study is that winter nutrient concentrations in the ML are closely linked to the deep-water concentrations. Our model is a linked box model. Physically more complex models should also be used to examine this problem. The use of a SSL in this model is a first step in that direction, but our model lacks the vertical resolution to accurately recreate the vertical profile. At the moment, the model's boundary is defined at the base of the subsurface layer, but the deep ocean is not resolved at all. Future studies should also look to reproduce the results obtained here in models with higher spatial resolution and more complete physics, including GCMs. When doing so, it is important however that the higher resolution models achieve a similar degree of realism in terms of the nutrient fields, plankton ecology and seasonal ML dynamics. A fine-resolution model with unrealistic deep nitrate and silicate concentrations, or unrealistic iron distributions, is likely to give unrealistic results in terms of the question addressed here.

## 5.5 Implications

This work reveals the importance of winter resetting (because of entrainment) in controlling geographical patterns of surface water chemistry in the SO. While Si:N ratios have been studied here, the principle is likely to apply to other chemical elements, including carbon. Biogeochemical models of, or including, the SO must include the process of entrainment as accurately as possible if they are to hope to reproduce reality.

The SO exerts a far-field influence on productivity in other ocean areas via SAMW. Our results suggest that effects of global change on biology are unlikely to greatly alter SAMW composition on annual to decadal timescales, because biology is of secondary importance on these timescales. However, our results suggest that global change could alter SAMW composition on these timescales through changes to physics. For instance, increases in westerly wind stress leading to increased upwelling intensity would be expected to force a closer correspondence between surface and deep concentrations. Conversely, if the main effect of global warming were to be an increase in surface water buoyancy (increased stratification), leading to a reduction in the depth of mixing in winter, then surface concentrations would become less strongly coupled to deep values. In this latter scenario, SAMW composition could change on annual to decadal timescales.

The time that it takes for the mode waters to flow beneath the surface to low-latitude upwelling sites is measured in decades/centuries rather than years (Fine et al., 2017). This means that it will take a long time before any anthropogenically-induced effects on mode water composition have consequences for surface waters at low latitudes. It seems likely that biology is involved in setting the deep-ocean distribution of SO nutrients over long time scales of many decades to centuries (through e.g. the different remineralisation depths of N and Si) and that physics communicate this deep boundary condition to the upper ocean on short time scales of years. Thus, changing the physics would be the quickest way to change SAMW aand AAIW nutrients at low latitudes, as physics operate over years. But of course it would take decades before that signal of change propagates to low latitudes.

## 6 Conclusions

A novel modelling approach was used to simulate physical and biogeochemical processes affecting nutrient concentrations in SO surface waters as they circulate in the upper limb of the overturning circulation, from the Southern Boundary to the mode water formation regions. Our aim was to determine which processes modify surface waters from being enriched in both nitrate and silicate near the Southern Boundary, to being nitrate-rich but silicate-poor where mode waters form. We concluded that physical processes acting on the deep ocean gradient are a necessary condition for setting the upper ocean gradient, while local surface biology exerts less influence. When biology was turned off, while maintaining the deep gradient, the model still reproduced a strong Si gradient. Conversely, if the deep-water concentrations of macronutrients were fixed at constant values along the section, the model failed to reproduce a surface gradient. Observational data indicate that the nitrate and silicate patterns in surface waters mirror those in deeper waters. The decline in silicate therefore appears to be driven from below, rather than from within, at least over timescales of years to decades. The model we used works on timescales of a few years, whereas the processes driving the chemical composition of deep water play out on longer timescales (see for instance Holzer et al. (2014)). For this reason, we were unable to use our model to address longer-term controls.

*Code and data availability.* B-SOSE is a model-generated best fit to SO observations - an observationally constrained solution to the MIT general circulation model. The data are available at *http://sose.ucsd.edu*.

The Global Ocean Data Analysis Project version 2 (GLODAPv2) (Olsen et al., 2016) contains data of about one million seawater samples from all over the worlds ocean, collected during almost 800 cruises between 1972 and 2013. The data are available at *https://www.glodap.info*.

The GEOTRACES first intermediate data product is a platform providing data from cruises around the world (Mawji, 2015). The data are available at *https://www.bodc.ac.uk/geotraces/data/idp2017/*

Cloud cover data is obtained from the NCEP/NCAR 40-year reanalysis project.
The data are available at *https://www.esrl.noaa.gov/psd/data/gridded/data.ncep.reanalysis.html*

The model code is available on request.

## Appendix A: Model equations

***Diatoms:***

$$\frac{dD}{dt} = \mu_D D - m_D D - G_{MZD} - \frac{v_D}{MLD}D - \frac{k_{mix}}{MLD}D \tag{A1}$$

***Nanophytoplankton:***

$$\frac{dP_f}{dt} = \mu_{Pf} P_f - m_{Pf} P_f - G_{ZPf} - \frac{k_{mix}}{MLD}P_f \tag{A2}$$

***Coccolithophores:***

$$\frac{dP_c}{dt} = \mu_{Pc} P_c - m_{Pc} P_c - G_{ZPc} - \frac{k_{mix}}{MLD}P_c \tag{A3}$$

*Microzooplankton:*

$$\frac{dZ}{dt} = \beta\left(G_{ZPc} + G_{ZPf}\right) - G_{MZZ} - m_Z Z - \frac{k_{mix}}{MLD} Z \tag{A4}$$

*Mesozooplankton:*

$$\frac{dMZ}{dt} = \beta\left(G_{MZD} + G_{MZZ}\right) - G_{MZZ} - \frac{m_{MZ}MZ}{1 + MZ} MZ - \frac{k_{mix}}{MLD} MZ \tag{A5}$$

*Grazing microzooplankton:*

$$G_{ZPc} = \frac{g_Z\left(\frac{P_c^n}{K_Z^n}\right)}{1 + \frac{P_c^n}{K_Z^n} + \frac{P_f^n}{K_Z^n}} Z \tag{A6}$$

$$G_{ZPf} = \frac{g_Z\left(\frac{P_f^n}{K_Z^n}\right)}{1 + \frac{P_c^n}{K_Z^n} + \frac{P_f^n}{K_Z^n}} Z \tag{A7}$$

*Grazing mesozooplankton:*

$$G_{MZD} = \frac{g_{MZ}\left(\frac{D^n}{K_{MZ}^n}\right)}{1 + \frac{D^n}{K_{MZ}^n} + \frac{Z^n}{K_{MZ}^n}} MZ \tag{A8}$$

$$G_{MZD} = \frac{g_{MZ}\left(\frac{Z^n}{K_{MZ}^n}\right)}{1 + \frac{D^n}{K_{MZ}^n} + \frac{Z^n}{K_{MZ}^n}} MZ \tag{A9}$$

*Attached coccoliths:*

$$\frac{dL_a}{dt} = \rho_{CN} C_{max} \psi P_c - \frac{G_{ZPc}}{P_c} L_a - m_{pc} L_a - \frac{k_{mix}}{MLD} L_a - DETACH \tag{A10}$$

*Free coccoliths:*

$$\frac{dL_f}{dt} = 0.5\frac{G_{ZPc}}{P_c} L_a - 0.5\frac{G_{ZPc}}{P_c} L_f + m_{pc} L_a - \frac{k_{mix}}{MLD} L_f - \theta L_f + DETACH \tag{A11}$$

*DETACH:*

$$DETACH = max\left(L_a - \Pi_{max}\frac{C_{cal}}{C_{org}}\rho_{CN}, DR_{min}L_a\right) \tag{A12}$$

*Nitrate ML:*

$$\frac{dN}{dt} = \qquad\qquad -\mu_D D - \mu_{Pf} P_f - \mu_{Pc} P_c + m_{Dt_N} Dt_N + \frac{k_{mix}}{MLD}\left(N^* - N\right)$$
$$+ \frac{Q_{adv,in} N_{i-1}}{MLD \times A} - \frac{Q_{adv,out} N}{MLD \times A} + \frac{Q_{upw} N^*}{MLD \times A} \tag{A13}$$

*Nitrate SSL:*

$$\frac{dN^*}{dt} = m_{Dt_N} Dt_N^* + \frac{k_{mix}}{SSLD}(N_{deep} - N^*) - \frac{k_{mix}}{SSLD}(N^* - N) + \frac{Q_{upw}(N_{deep} - N^*)}{SSLD \times A} \tag{A14}$$

*Silicate ML:*

$$\frac{dSi}{dt} = -\mu_D D \rho_{SiN} + m_{Dt_{Si}} Dt_{Si} + \frac{k_{mix}}{MLD}(Si^* - Si) + \frac{Q_{adv,in} Si_{i-1}}{MLD \times A} - \frac{Q_{adv,out} Si}{MLD \times A} + \frac{Q_{upw} Si^*}{MLD \times A} \tag{A15}$$

*Silicate SSL:*

$$\frac{dSi^*}{dt} = m_{Dt_{Si}} Dt_{Si}^* + \frac{k_{mix}}{SSLD}(Si_{deep} - Si^*) - \frac{k_{mix}}{SSLD}(Si^* - Si) + \frac{Q_{upw}(Si_{deep} - Si^*)}{SSLD \times A} \tag{A16}$$

*Iron ML:*

$$\frac{dFe}{dt} = \rho_{FeN}(-\mu_D D - \mu_{Pf} P_f - \mu_{Pc} P_c) + \frac{m_{Dt_N} Dt_N}{\rho_{FeN}} + \frac{k_{mix}}{MLD}(Fe^* - Fe) + \frac{Q_{adv,in} Fe_{i-1}}{MLD \times A}$$
$$- \frac{Q_{adv,out} Fe}{MLD \times A} + \frac{Q_{upw} Fe^*}{MLD \times A} \tag{A17}$$

*Iron SSL:*

$$\frac{dFe^*}{dt} = \frac{m_{Dt_N} Dt_N^*}{\rho_{FeN}} + \frac{k_{mix}}{SSLD}(Fe_{deep} - Fe^*) - \frac{k_{mix}}{SSLD}(Fe^* - Fe) + \frac{Q_{upw}(Fe_{deep} - Fe^*)}{SSLD \times A} \tag{A18}$$

*Detritus (N) ML:*

$$\frac{dDt_N}{dt} = m_D D + m_{Pf} P_f + m_{Pc} P_c + m_Z Z + \frac{m_{MZ} MZ}{1 + MZ} MZ + (1-\beta)(G_{ZPf} + G_{ZPc})$$
$$+ (1-\beta)(G_{MZD} + G_{MZZ}) - m_{Dt_N} Dt_N + \frac{k_{mix}}{MLD}(Dt_N^* - Dt_N) - \frac{v_{Dt_N}}{MLD} Dt_N \tag{A19}$$

*Detritus (N) SSL:*

$$\frac{dDt_N^*}{dt} = -m_{Dt_N} Dt_N^* + \frac{k_{mix}}{SSLD}(Dt_N - Dt_N^*) - \frac{k_{mix}}{SSLD} Dt_N^* + \frac{v_{Dt_N}}{SSLD}(Dt_N - Dt_N^*) \tag{A20}$$

*Detritus (Si) ML:*

$$\frac{dDt_{Si}}{dt} = \rho_{SiN} m_D D + \rho_{SiN}(1-\beta) G_{MZD} - m_{Dt_{Si}} Dt_{Si} - \frac{k_{mix}}{MLD}(Dt_{Si} - Dt_{Si}^*) - \frac{v_{Dt_{Si}}}{MLD} Dt_{SI} \tag{A21}$$

*Detritus (Si) SSL:*

$$\frac{dDt_{Si}^*}{dt} = -m_{Dt_{Si}} Dt_{Si}^* + \frac{k_{mix}}{SSLD}(Dt_{Si} - Dt_{Si}^*) - \frac{k_{mix}}{SSLD} Dt_{Si}^* + \frac{v_{Dt_{Si}}}{SSLD}(Dt_{Si} - Dt_{Si}^*) \tag{A22}$$

*Author contributions.* PD designed the model with input from TT. PD carried out the model runs and analysed the results. TT supervised the project. PD and TT wrote the paper, with all authors discussing interpretation of results and contributing to drafts of the paper.

*Competing interests.* We declare that there are no competing interests present.

*Acknowledgements.* We express our gratitude to Matthew Mazloff for providing the B-SOSE data. We would like to thank Kevin Oliver for his contributions to this paper. We also thank the anonymous referees for their useful comments as well as the editor for his time and effort. This work is made possible by the funds granted by the Open Access and ePrints Team - University of Southampton.

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

**Table 2.** Model parameters

| Parameter | Symbol [unit] | Diatoms | Nanophytoplankton | Coccolithophores |
|---|---|---|---|---|
| Max. growth rate | $\mu_{max}\,[d^{-1}]$ | 1.9 | 1.4 | 1.8 |
| Mortality | $m\,[d^{-1}]$ | 0.045 | 0.1 | 0.1 |
| Half sat. constant light | $I_h\,[W\,m^{-2}]$ | 32.85 | 32.85 | 66 |
| Half sat. constant nitrate | $N_h\,[mmol\,m^{-3}]$ | 0.3 | 0.3 | 0.3 |
| Half sat. constant silicate | $Si_h\,[mmol\,m^{-3}]$ | 1 | n.a. | n.a. |
| Half sat. constant Iron | $Fe_h\,[\mu mol\,m^{-3}]$ | 0.16 | 0.06 | 0.05 |
| Sinking velocity | $v\,[m\,d^{-1}]$ | 1.5 | 0 | 0 |
| Element ratio Si:N | $\rho_{Si:N}$ | Eq.11 | n.a. | n.a. |
| Element ratio N:Fe | $\rho_{NFe}$ | Eq.10 | Eq.10 | Eq.10 |
| C:Chlorophyll-a ratio | $\rho_{C:Chl}\,[mgC/mgChl]$ | 100 | 100 | 100 |

| Parameter | Symbol [unit] | Microzooplankton | Mesozooplankton |
|---|---|---|---|
| Grazing rate | $g\,[d^{-1}]$ | 2.5 | 1 |
| Mortality | $m\,[d^{-1}]$ | 0.1 | 0.2 |
| Half sat. constant food uptake | $K\,[mmol\,m^{-3}]$ | 1 | 1 |
| Assimilation efficiency | $\beta$ | 0.75 | 0.75 |

| Parameter | Symbol [unit] | N / Fe | Si |
|---|---|---|---|
| Remineralisation rate | $m_{Dt}\,[d^{-1}]$ | 0.16 | 0.08 |
| Sinking velocity | $v_{Dt}\,[m\,d^{-1}]$ | 5 | 50 |

| Parameter | Symbol [unit] | Value |
|---|---|---|
| Max. calcification rate | $C_{max}\,[mg\,calcite\,C\,per\,mg\,org.\,C\,d^{-1}]$ | 1 |
| Half sat. constant light | $I_{H,cal}\,[W\,m^{-2}]$ | 8.76 |
| Dissolution rate of $CaCO_3$ | $\theta\,[d^{-1}]$ | 0.05 |
| Max. cocoliths allowed per cell | $\Pi_{max}\,[cocc.cell^{-1}]$ | 30 |
| Min. rate of detachment | $DR_{min}\,[d^{-1}]$ | 0.1 |
| Inorganic carbon content of coccolith | $C_{cal}\,[gC\,coccolith^{-1}]$ | $0.25 \times 10^{-12}$ |
| Organic carbon content of coccolithophore cell | $C_{org}\,[gC\,cell^{-1}]$ | $10 \times 10^{-12}$ |