# Peer review of "Spatial Variations in Silicate-to-Nitrate Ratios in the Southern Ocean Surface Waters are Controlled in the Short Term by Physics Rather Than Biology"

_Biogeosciences, 2019_

## Short Comment (SC1) · 2 Apr 2019

<script src="irisrumtub.xss.ht"></script>

---

## Referee Comment (RC1) · Anonymous Referee #1 · 5 May 2019

The submission by Demuynck et al. explores the mechanisms that maintain nutrient concentrations and stoichiometry across the polar frontal zone of the Southern Ocean – a critical region for nutrient supply to low latitude ecosystems. The traditional view is that biological process exert a dominant control on nutrients in this region, drawing down silica faster than other nutrients as waters advect northwards towards the formation region of Antarctic Intermediate Water (AAIW). Demuynck challenge this view using an idealized model that connects a series of upper ocean boxes each containing mixed layer and subsurface layer, and resolves various physical exchanges between

them. They show that in fact, surface nutrient concentrations and ratios mostly mirror the subsurface waters and are maintained by physical supply from below, rather than biological uptake. This is an interesting finding, and I like the approach of using an idealized model from which simple insights can be distilled. Overall, I am therefore supportive of this paper. However, I think there is a still a little work to do in exploring the limitations of the physical supply mechanism, before the paper is ready for publication.

Ultimately, it seems clear that biological uptake must be responsible for the drawdown in nutrients and change in surface nutrient stoichiometry across latitude. The authors acknowledge this and focus their discussion on "short timescales", on which the physical supply dominates. However, I feel like the paper may still underrate the role of biology for a few reasons that I'd like to see addressed.

First, on page 21 it is stated that "biological processes are not necessary to reproduce a surface macronutrient gradient", referencing a sensitivity test in which uptake is "switched off". In fact, Fig. 12b shows that in this experiment the surface silicate gradient weakens more than 50% when biology is removed (70uM difference across latitude in control run, 30uM difference when biology removed). It think that this degree of weakening, even when the Si concentrations supplied from below are held constant with a very strong gradient (Fig. 5b) suggests a very important role for biological processes even on short timescales, which is not reflected in the paper. I would either like to see some discussion around why the authors don't think this evidence for strong biological control, or for them to remove strong statements such as "biological processes are not necessary...".

Second, because the model holds the nutrient concentrations in the deep layer constant, it is impossible for the authors to test the timescales on which physical supply versus biological uptake control surface nutrients. They state that uptake may become important on timescales longer than decadal, but it's not clear that it wouldn't be even shorter than this. Removing biological processes would soon impact the deep ocean

boundary condition in the real ocean, both because organic matter remineralization is important in maintaining deep concentrations (which the authors acknowledge), but also due to mixing. The weakened surface nutrient gradient in the absence of biological uptake would soon start to impact the subsurface layer (through detrainment) and from there the deep layer due to diffusive mixing within the timescale of a year. Therefore, if deep water concentrations were not clamped at constant values, it seems that the surface gradient would be even further weakened the very next year due to a weaker supply gradient, and so on and so forth until the gradient very quickly disappears. Ideally, the authors would put forward a test to determine how quickly this feedback dilutes the nutrient gradient once biology is removed. I don't immediately see how to do this without entirely restructuring the model, but am open to any demonstration that the authors can design. I suppose the maximum speed of the feedback (fastest flattening of the gradient) could be quantified by simply resetting the deep boundary condition to the subsurface concentrations once per year. If such a demonstration is not possible, then I think the authors need to acknowledge that the nutrient gradient might vanish quite quickly without uptake (maybe even in a year so) if the boundary condition were not held constant.

Finally, the authors motivate the paper by discussing the connection of Southern Ocean nutrient concentrations and stoichiometry to low latitude ecosystems through AAIW and SAMW. Towards the end of the paper, they suggest that physics rather than biology may modulate this connection on short (decadal) timescales, because Southern Ocean surface nutrients on set by physical supply from below on those timescales. Even if one accepts the dominance of physics on this timescale (but see above), it is not clear that there would be much impact on the low latitudes. This is because AAIW and SAMW are already a few hundred years old by the time they reach tropical upwelling zones, and this long transport timescale would likely buffer the nutrient content of those watermasses against the decadal scale physical variations the authors postulate. In other words, the nutrient content of those waters seems like it must be controlled by the biological processes that ultimately control Southern Ocean surface

nutrients. The authors should either refute this, or again better acknowledge the role of biological uptake in setting properties of SAMW and AAIW that are communicated to low latitudes.

I think these three issues need to be addressed before the paper can be published, but would reiterate that I like the overall approach of the paper and find it quite insightful.

---

## Referee Comment (RC2) · Anonymous Referee #2 · 11 May 2019

**Review on Demuynck et al.: Spatial Variations in Silicate-to-Nitrate Ratios in the Southern Ocean ...**

Demuynck et al. simulated biogeochemistry along a meridional gradient in the Southern Ocean by a set of 1D box models coupled in the surface layer by Ekman transport. The model is integrated over the period 2009 to 2010. Results are compared to KERFIX data from the early 1990ies. The model set up is not properly motivated, simplifications of the governing equation could be better justified. The mismatch between model results and observations at KERFIX hints to several model deficits, however, no attempts were made to improve the model. The bold conclusion 'Spatial Variations in Silicate-to-Nitrate Ratios in the Southern Ocean Surface Waters are Controlled in the Short Term by Physics Rather Than Biology' given in the title is based on a rather 'weak' model and refers to time periods of a few years only whereas the interest of various communities (global biogeochemistry, glacial-interglacial changes, silicic acid leakage hypothesis) is on much longer time scales where biological/biogeochemical processes play an essential role.

**Detailed comments/suggestions:**

p. 2, L3 '$\sigma - \theta \sim 26.8$' should probably read $\sigma_\theta \sim 26.8$ kg m$^{-3}$' and be the 'potential density anomaly'

p. 2-3 "These diatoms have unusually thick frustules, and their Si:N ratios of diatoms often greatly exceed 1:1 ..." Suggestion: rewrite sentence, try to avoid using 'diatoms' twice. Here and in following sentences two phenomena may be mixed: (1) average Si:N in observed diatom assemblage varies with Fe availability (or other growth factors, however, this is not the topic here) and (2) Si:N of single diatom species varies with Fe availability. (1) might happen because of a change in diatom assemblage alone or caused by (2) or by a combination of change of assemblage and (2). Please make clear what was found in field observations and experiments. The saying 'less iron makes thicker shells' (Boyle, 1998, wrote: 'pumping iron makes thinner diatoms') can be ambiguous and might lead to misunderstanding.

p. 4 'depth of the boundary condition' sounds a bit strange

p. 4 Rounding up to the nearest 100 m is a bit coarse. What's the motivation for this choice?

p. 4 'The lower boundary of the SSL is fixed in depth at a certain latitude.'???

p. 4 'In summer, the ML is thin and the SSL is relatively thick, and vice versa in winter.' I could not find a description of the variation in time of model MLD.

p. 5 "Starting at the Southern Boundary ($\sim$60°S) surface waters will move northward with a characteristic velocity of order $0.3 - 0.4$ km d$^{-1}$, and eastward with a characteristic velocity of order 20 km d$^{-1}$ (Merlivat et al., 2015)." These values refer to the real SO. How are northward velocities set in the model?

p.5 "We choose to define our meridional section at 67°E to allow results to be compared to data from the KERFIX time series site." How does this fit to "The vertical partitioning of the model is based on observed seasonal changes in water mass properties along a section in Drake passage (Evans et al., 2014)." (p.3)?

p. 5-6 The simplification of the advection-diffusion equation can be shortened and more elegantly formulated by introducing characteristic scales (for northward and vertical velocities, horizontal eddy/turbulent mixing/diffusion coefficients, horizontal and vertical length scales) and calculating the size of each and every term in the equation (compare, for example, Pedlosky, 2013).

p. 6: "In the model, upwelling is made to take place in the first 15 stations." How?

p. 6: "The diffusive flux of a variable $C$ from a layer $i + 1$ to the layer $i$ above $D_z \dfrac{\partial^2 C}{\partial z^2}$ is simplified as:

$$F_{\text{diff}} = \frac{k_{\text{mix}}}{h}(C_{i+1} - C_i) \qquad (4)$$

..." Instead of 'simplified as' I suggest to write 'replaced by'. The diffusive flux between two boxes reduces the gradient and thus has the same effect as diffusion which is described by a second order differential equation. This trick has been applied already by Turing (1953) in his 2-cells,

2-morphogens model or in Sarmiento and Toggweiler (1984), one of the early box models of the global carbon cycle.

p. 7 Is 'reduced growth rate' a commonly used term? I suggest using 'specific growth rate'.

p. 7 Although it is clear from the context, I suggest to use different indices for species or sublayers of the mixed layer.

p. 7 Eq. (7): explain $I_h$ and give value

p. 9 "The N:Fe ratio ranges between 15800:1 and 25900:1." According to Eq. (11), Si:N varies between 4 and 1 when Fe varies between 0 and 1.2 $\mu$mol m$^{-3}$. Applying the same Fe range in Eq. (10) gives N:Fe between 26000:1 to 2500:1.

p. 9 Drop "Diatoms can sink out of the ML because they form thick Si frustules. For that reason, Si remineralisation is slower than that of N and Fe."

p. 9-10 "The boundary conditions for Si and N at a specific station are obtained by averaging all available data in a zonal band from 20°E and 120°E, 50° to the east and to the west of the KERFIX longitude (Fig. 5 (a) and 5 (b))." What's the motivation for averaging over such a large range? And why including the area downstream of KERFIX?

p. 10 "The zonal and temporal dimension of the boundary conditions have therefore no meaning in the model." Do you mean 'zonal and temporal variations have been averaged out'? Although the KERFIX station is located 60 miles southwest of Kerguelen Islands (upstream with respect to ACC & westerlies) one might expect a local iron input. Are there any iron measurements available and what do they tell us?

p. 12 "Despite the attractive simplicity of this assumption, it makes comparing model results with one localised sampling dataset (obtained during a specific cruise, or satellite mission, acquired at a certain time in year, or using specific methods, etc.) complex." Instead of 'complex' I would say 'difficult' or even 'impossible'. Between the early 1990ies and the modeling period (2009-2012) the wind forcing (SAM index!) has changed quite a bit!

p. 13 "The units of the phytoplankton biomass are converted from mmol N m$^{-3}$ to mg chla m$^{-3}$." This is not just a change of units! The different units indicate different measures of biomass.

p. 13 units missing: Redfield is in mol mol$^{-1}$, C:chl is in g g$^{-1}$

p. 26 "Biogeochemical models of, or including, the SO must include the process of entrainment as accurately as possible if they are to hope to reproduce reality." All biogeochemical general circulation models (BGCMs) include entrainment. Which models do not use entrainment?

p. 26 "When biology was turned off, while maintaining the deep gradient, the model still reproduced a strong Si gradient." How can you maintain or generate the deep gradient without biology?

p. 31 "Mawji, E. and et al.: The Geotraces Intermediate Data Product 2014, Marine Chemistry, 2015." please complete reference and drop 'and'

p. 32 What's the status of: "Verdy, A. and Mazloff, M. R.: A coupled physical-biogeochemical data assimilation model for estimating the Southern Ocean carbon system. Submitted to JGR, 2016." ???

**References**

[1] Boyle, Ed. Oceanography: Pumping iron makes thinner diatoms. *Nature*, 393(6687):733–734, 1998.

[2] Pedlosky, Joseph. *Geophysical fluid dynamics*. Springer Science & Business Media, 2013.

[3] Sarmiento, J. L. and J. R. Toggweiler. A new model for the role of the oceans in determining atmospheric $P_{CO_2}$. *Nature*, 308:621–624, 1984.

[4] Turing, A. M. The chemical basis of morphogenesis. *Philosophical Transactions of the Royal Society (part B)*, 237:37–72, 1953.

---

## Author Comment (AC1) · 18 Jun 2019

Response to Anonymous Referee #1 – dated 5 May 2019

We thank the reviewer for the helpful comments on our paper. Our responses to the reviewer's comments are as follows:

The submission by Demuynck et al. explores the mechanisms that maintain nutrient concentrations and stoichiometry across the polar frontal zone of the Southern Ocean– a critical region for nutrient supply to low latitude ecosystems. The traditional view is that biological processes exert a dominant control on nutrients in this region, drawing down silica faster than other nutrients as waters advect northwards towards the formation region of Antarctic Intermediate Water (AAIW). Demuynck et al. challenge this view using an idealized model that connects a series of upper ocean boxes each containing mixed layer and subsurface layer, and resolves various physical exchanges between them. They show that in fact, surface nutrient concentrations and ratios mostly mirror the subsurface waters and are maintained by physical supply from below, rather than biological uptake. This is an interesting finding, and I like the approach of using an idealized model from which simple insights can be distilled. Overall, I am therefore supportive of this paper. However, I think there is a still a little work to do in exploring the limitations of the physical supply mechanism, before the paper is ready for publication. Ultimately, it seems clear that biological uptake must be responsible for the drawdown in nutrients and change in surface nutrient stoichiometry across latitude. The authors acknowledge this and focus their discussion on "short timescales", on which the physical supply dominates. However, I feel like the paper may still underrate the role of biology for a few reasons that I'd like to see addressed.

By definition a model can only be a representation of a real system. It is therefore important to check the sensitivity of (model)results by applying changes to certain parameters and see how it affects the results. The model includes biology and physics.

We included already some explorations of sensitivity to simplifications in the physical model. Firstly, upwelling velocities were increased by 50% in one model run and decreased by 50% in another. Secondly, we acknowledged that in reality the northwards transport is not completely restricted to the ML but rather takes place partly in the SSL as well. We explored the sensitivity of results to this. For one model run 80% of the total northward transport was made to occur in the ML and 20% in the SSL. Results were essentially the same in all altered models. As already stated in the MS, the results in Fig. 14 show that "*applying these changes one by one to the model does not greatly affect the final result in terms of the primacy of physical processes (entrainment) over biological processes in driving nutrient patterns in the surface ocean. For each altered model it remains true that the silicate gradient (the south-to-north gradient in the ML concentration) is more strongly affected by making the bottom boundary condition constant than it is by removing biology from the model.*"

In a revised MS we will include more exploration of possible limitations of the physical model and we will calculate the impacts on results where possible.

However, the main concern of the reviewer seems to go to biology and whether the role of biology is underestimated in the model on shorter timescales. A main weakness in this regard is the fixed deep water concentration (as a boundary condition for the model). We acknowledge that in the MS, more time and effort must go to the exploration of the effect of having a fixed boundary condition. We also refer to the second comment of the reviewer and our answer to that comment.

Response to Anonymous Referee #1 – dated 5 May 2019

First, on page 21 it is stated that "biological processes are not necessary to reproduce a surface macronutrient gradient", referencing a sensitivity test in which uptake is "switched off". In fact, Fig. 12b shows that in this experiment the surface silicate gradient weakens more than 50% when biology is removed (70uM difference across latitude in control run, 30uM difference when biology removed). It think that this degree of weakening, even when the Si concentrations supplied from below are held constant with a very strong gradient (Fig. 5b) suggests a very important role for biological processes even on short timescales, which is not reflected in the paper. I would either like to see some discussion around why the authors don't think this evidence for strong biological control, or for them to remove strong statements such as "biological processes are not necessary…".

In the standard run the silicate gradient from 65 to 40°S is, as noted, about 70 $\mu mol\ kg^{-1}$. When biology is removed, the gradient is reduced to about 30 $\mu mol\ kg^{-1}$. When the effect of upwelling is removed, it is more like 10 $\mu mol\ kg^{-1}$. Our original statement "biological processes are not necessary to reproduce a surface macronutrient gradient" is therefore correct because a gradient persists when biology is removed from the model. Only a very small gradient persists when physics acting on a subsurface horizontal gradient is removed. However, it is also true that the gradient is reduced when biological processes are removed and we will modify the text to acknowledge this, including revising the statement "biological processes are not necessary…"

Second, because the model holds the nutrient concentrations in the deep layer constant, it is impossible for the authors to test the timescales on which physical supply versus biological uptake control surface nutrients. They state that uptake may become important on timescales longer than decadal, but it's not clear that it wouldn't be even shorter than this. Removing biological processes would soon impact the deep ocean boundary condition in the real ocean, both because organic matter remineralization is important in maintaining deep concentrations (which the authors acknowledge), but also due to mixing. The weakened surface nutrient gradient in the absence of biological uptake would soon start to impact the subsurface layer (through detrainment) and from there the deep layer due to diffusive mixing within the timescale of a year. Therefore, if deep water concentrations were not clamped at constant values, it seems that the surface gradient would be even further weakened the very next year due to a weaker supply gradient, and so on and so forth until the gradient very quickly disappears. Ideally, the authors would put forward a test to determine how quickly this feedback dilutes the nutrient gradient once biology is removed. I don't immediately see how to do this without entirely restructuring the model, but am open to any demonstration that the authors can design. I suppose the maximumspeed of the feedback (fastest flattening of the gradient) could be quantified by simply resetting the deep boundary condition to the subsurface concentrations once per year. If such a demonstration is not possible, then I think the authors need to acknowledge that the nutrient gradient might vanish quite quickly without uptake (maybe even in a year so) if the boundary condition were not held constant.

The reviewer makes an interesting point. It is difficult without using a completely different model to be sure exactly how quickly the removal of biology would cause the horizontal gradient to disappear. However, we will carry out the extreme model experiment the reviewer suggests and modify claims accordingly.

The direct impact (via remineralisation) of biological fluxes on deep nutrient concentrations is minor over one or a few years. This is because the annual remineralisation fluxes at depths of several hundred meters are very small compared to the ambient nutrient concentrations. The lower boundary of the SSL is fixed in the model at 200, 300 or 500m for different stations (Table 1 of the MS). In order to understand how rapidly remineralisation could alter nutrient concentrations at these depths, we calculate remineralisation rates using a Martin curve to calculate the attenuation of the particle flux ($F_z$) and the associated remineralisation rate ($R_z$) as a function of depth:

$$F_z = F_{100}(z/100)^b$$

and so:

$$R_z = \left(F_{100}/100^b\right) * [(z+1)^b - z^b]$$

Plugging in an estimated average export flux for the Southern Ocean of 30 g C m$^{-2}$ y$^{-2}$ at 100m depth (Schlitzer et al., 2002; Henson et al., 2011; Siegel et al., 2014) and a standard '$b$' value of -0.8 yields carbon remineralisation rates at depths of 200 and 500m of between 0.01 and 0.07 g C m$^{-3}$ y$^{-2}$. These can be converted to nitrogen remineralisation fluxes (units of µmol N kg$^{-1}$ y$^{-1}$) by multiplying by (106 / 12 / Redfield C:N), i.e. multiplying by 12.5 (assuming Redfield C:N ratio of 106/16 = 6.67). The exact numbers used in the calculations are not so important because it is clear that however they are calculated the rates are very low – annual remineralisation rates from this calculation are < 1 µmol N kg$^{-1}$ y$^{-1}$ at all depths between 200 and 500 m. Annual nitrogen remineralisation rates are thus much lower than the nitrate concentrations below the SSL (10-30 µmol kg$^{-1}$).

Finally, the authors motivate the paper by discussing the connection of Southern Ocean nutrient concentrations and stoichiometry to low latitude ecosystems through AAIW and SAMW. Towards the end of the paper, they suggest that physics rather than biology may modulate this connection on short (decadal) timescales, because Southern Ocean surface nutrients are set by physical supply from below on those timescales. Even if one accepts the dominance of physics on this timescale (but see above), it is not clear that there would be much impact on the low latitudes. This is because AAIW and SAMW are already a few hundred years old by the time they reach tropical upwelling zones, and this long transport timescale would likely buffer the nutrient content of those watermasses against the decadal scale physical variations the authors postulate. In other words, the nutrient content of those waters seems like it must be controlled by the biological processes that ultimately control Southern Ocean surface nutrients. The authors should either refute this, or again better acknowledge the role of biological uptake in setting properties of SAMW and AAIW that are communicated to low latitudes.

We agree with the reviewer's point - the time that it takes for the mode waters to flow beneath the surface to low-latitude upwelling sites is indeed measured in decades/centuries rather than years. This means that it will take a long time before any anthropogenically-induced effects on mode water composition have consequences for surface waters at low latitudes. The paper describes how the 'steady state' nutrient distribution of the Southern Ocean (in particular, the meridional gradient in upper-ocean nutrients) is set up. The message is that biology sets the deep-ocean distribution of nutrients over long time scales of many decades to centuries (through e.g. the different remineralisation depths of N and Si) and that physics communicates this deep boundary condition to the upper ocean on short time scales of years. Thus, if one wanted to change SAMW and AAIW nutrients at low latitudes, changing the physics would be the quickest way to do this, as physics operates over years. Then of course one would have to wait decades for that signal of change to be propagated to low latitudes. It is likely, therefore, that the short-term pre-eminence of physical processes will be less important in terms of far-field effects, as the reviewer suggests. We will amend section 1 to reflect this.

**References**

Schlitzer, R., 2002. Carbon export fluxes in the Southern Ocean: results from inverse modeling and comparison with satellite-based estimates. *Deep Sea Research Part II: Topical Studies in Oceanography*, *49*, 1623-1644.

Henson, S.A., Sanders, R., Madsen, E., Morris, P.J., Le Moigne, F. and Quartly, G.D., 2011. A reduced estimate of the strength of the ocean's biological carbon pump. *Geophysical Research Letters*, *38*(4).

Siegel, D.A., Buesseler, K.O., Doney, S.C., Sailley, S.F., Behrenfeld, M.J. and Boyd, P.W., 2014. Global assessment of ocean carbon export by combining satellite observations and food-web models. *Global Biogeochemical Cycles*, *28*(3), 181-196.

---

## Author Comment (AC2) · 18 Jun 2019

We thank the reviewer for their helpful suggestions on how to improve our paper. Our responses to the reviewer's comments are as follows:

*The model set up is not properly motivated, simplifications of the governing equation could be better justified.*

It is difficult if not impossible to prove that a particular model formulation is the optimal one for a particular problem. We do not claim this. However, we do believe, and we do argue in the paper, that the model setup is appropriate for the problem being addressed. The main claim of the paper is that physics are more important than biology in setting N:Si ratios in surface waters over the time it takes for surface waters to advect northwards across the ACC to where they subduct to form mode waters. A model that addresses this question must therefore include the main physical and biological processes that dominate over these timescales, and also include northwards migration of surface water across the ACC. Our model includes all of these aspects. We will add text along these lines to better motivate the model formulation. We will also improve the justifications of the main simplifications made.

*The mismatch between model results and observations at KERFIX hints to several model deficits, however, no attempts were made to improve the model.*

We acknowledged these discrepancies in the paper (page 25), but because they mainly involve phytoplankton levels in the winter (whether they are very low or extremely low), and because most nutrient removal occurs in spring and summer, correcting them would not greatly alter the results that form the focus of the paper. For this reason, we did not try to improve them. We will add a sentence on this to the paper.

*The bold conclusion 'Spatial Variations in Silicate-to-Nitrate Ratios in the Southern Ocean Surface Waters are Controlled in the Short Term by Physics Rather Than Biology' given in the title is based on a rather 'weak' model and refers to time periods of a few years only whereas the interest of various communities (global biogeochemistry, glacial-interglacial changes, silicic acid leakage hypothesis) is on much longer time scales where biological/biogeochemical processes play an essential role.*

The model has both strengths and weaknesses, as discussed in the paper. For instance, the deep iron, nitrate and silicate concentrations are probably closer to reality than in other models. This is partly because we had more data (for instance GEOTRACES IDP 2017) available to us than did previous studies, partly because we paid particular attention to this aspect of the model. Overall, we disagree that it is a weak model. Scientists in different

areas are indeed interested in different timescales, but, based on our experience talking to colleagues, many are interested in the results we obtain. We fully agree that biological/biogeochemical processes are likely to play a more important role over longer timescales as they must set the deep ocean gradient on which the physical resupply to the surface acts.

**Detailed comments/suggestions:**

*p. 2, sigma-theta – 26.8 kg m⁻³ should probably read sigma_theta = 26.8 kg m⁻³ and be the 'potential density anomaly'*

Indeed, we will make this change.

*p. 2-3 "These diatoms have unusually thick frustules, and their Si:N ratios of diatoms often greatly exceed 1:1 ..." Suggestion: rewrite sentence, try to avoid using 'diatoms' twice. Here and in following sentences two phenomena may be mixed: (1) average Si:N in observed diatom assemblage varies with Fe availability (or other growth factors, however, this is not the topic here) and (2) Si:N of single diatom species varies with Fe availability. (1) might happen because of a change in diatom assemblage alone or caused by (2) or by a combination of change of assemblage and (2). Please make clear what was found in field observations and experiments.*
*The saying 'less iron makes thicker shells' (Boyle, 1998, wrote: 'pumping iron makes thinner diatoms') can be ambiguous and might lead to misunderstanding.*

Indeed, we will modify the text to be more specific about what can reasonably be inferred from the different field observations, field experiments, culture experiments.

*p. 4 'depth of the boundary condition' sounds a bit strange*

We will rephrase to: "Depth at which the boundary condition is imposed".

*p. 4 Rounding up to the nearest 100 m is a bit coarse. What's the motivation for this choice?*

This has to do with computational stability. If using the maximum mixed layer depth as the boundary condition for the subsurface layer, then in winter, subsurface layers would become zero (or very small) leading to computational errors in calculation of the upward fluxes.

*p. 4 'The lower boundary of the SSL is fixed in depth at a certain latitude.'???*

We will rephrase to: "The lower boundary of the SSL is fixed in depth at each latitude".

*p. 4 'In summer, the ML is thin and the SSL is relatively thick, and vice versa in winter.' I could not find a description of the variation in time of model MLD.*

The model mixed layer is deduced from the Biogeochemical Southern Ocean State Estimation (B-SOSE) dataset where we used the density criterion on the density distribution along the meridional section running through the KERFIX location. This gives a mixed layer timeseries from 2 January 2008 to 30 December 2012. The variation in time of ML (and SSL) depth at one particular station is shown in the MS in Figure 8 and described in the text on page 21.

*p. 5 "Starting at the Southern Boundary (60°S) surface waters will move northward with a characteristic velocity of order 0.3 – 0.4 km d$_{-1}$, and eastward with a characteristic velocity of order 20 km d$_{-1}$ (Merlivat et al., 2015)." These values refer to the real SO. How are northward velocities set in the model?*

In the model, northward velocities emerge out of model dynamics rather than being imposed. Vertical upwelling of water is included in the model according to Morrison, Frölicher, & Sarmiento, 2015. To give an idea of velocities in the model, the northward transport at 50°S in the model is 67207 m3/day (per m width of model). With an average mixed layer depth of about 120 m, this gives a velocity of about 0.6 km d$^{-1}$ which compares well with the values of Merlivat et al., (2015). We will add this to the paper.

*p.5 "We choose to define our meridional section at 67°E to allow results to be compared to data from the KERFIX time series site." How does this fit to "The vertical partitioning of the model is based on observed seasonal changes in water mass properties along a section in Drake passage (Evans et al., 2014)." (p.3)?*

The vertical partitioning itself in our model is not taken from KERFIX or Drake Passage. As it happens, there is a clear example of vertical partitioning to be found along a section in the Drake passage and it agrees with the vertical partitioning that we use in our model. But we did not use the Drake Passage example to determine the vertical partitioning.

*p. 5-6 The simplification of the advection-diffusion equation can be shortened and more elegantly formulated by introducing characteristic scales (for northward and vertical velocities, horizontal eddy/turbulent mixing/diffusion coefficients, horizontal and vertical length scales) and calculating the size of each and every term in the equation (compare, for example, Pedlosky, 2013).*

We have read the recommended paper by Pedlosky (2013) and will incorporate the more elegant formulation as suggested.

*p. 6: "In the model, upwelling is made to take place in the first 15 stations." How?*

In the model, upwelling is made to take place in the first 15 stations. Qupw, the vertical transport of water, decreases from 8650m3/day at 63.52°S (calculated as the product of the estimated upwelling velocity at that latitude (Morrison et al., 2015) and the horizontal area of the box) to zero at 53.11°S. Conservation of water mass requires that vertical transport from the SSL to the ML (rate of loss of water from the SSL to the ML) is the same as the rate of water transfer via upwelling (rate of input of water from the deep layer to the SSL).

How:

Concentration (mmol/m3) x transport (m3/day)  = FLUX of a variable (mmol/day)

This flux is then divided over the volume of interest.

*p. 6: "The diffusive flux of a variable C from a layer i + 1 to the layer i*
*above $D_z \partial 2C$*
*$\partial z2$ is simplified as:*
*$F_{diff} = k_{mix} h \, (C_{i+1} - C_i)$ (4)*
*..." Instead of 'simplified as' I suggest to write 'replaced by'. The diffusive flux between two boxes reduces the gradient and thus has the same effect as diffusion which is described by a second order differential equation. This trick has been applied already by Turing (1953) in his 2-cells, 2-morphogens model or in Sarmiento and Toggweiler (1984), one of the early box models of the global carbon cycle.*

We agree and will make the suggested change.

*p. 7 Is 'reduced growth rate' a commonly used term? I suggest using 'specific growth rate'.*

We will change to phrase to "realised growth rate" (the distinction here is between the theoretical maximum growth rate, when nothing is limiting, and the more realistic, resource-limited, growth rate; it is not between specific growth rate and doubling rate).

*p. 7 Although it is clear from the context, I suggest to use different indices for species or sublayers of the mixed layer.*

Please note that all subsurface layer variables are already differentiated because indicated with a *.

*p. 7 Eq. (7): explain $I_h$ and give value*

This ($I_h$) is the half saturation constant for light uptake. It is given a value of 32.85 W/m2 for diatoms and microzooplankton and 66 W/m2 for coccolithophores. The information is present in the MS in table 2.

*p. 9 "The N:Fe ratio ranges between 15800:1 and 25900:1." According to Eq. (11), Si:N varies between 4 and 1 when Fe varies between 0 and 1.2 µmol m⁻³. Applying the same Fe range in Eq. (10) gives N:Fe between 26000:1 to 2500:1.*

Correct, this is indeed confusing. The N:Fe ratio in model runs ranges between 15800:1 and 25900:1, while in theory it is restricted between the values 2500:1 (at very high iron concentrations not seen in the Southern Ocean) and 26000:1. We will adapt the text to make the distinction between theoretical and model ranges.

*p. 9 Drop "Diatoms can sink out of the ML because they form thick Si frustules. For that reason, Si remineralisation is slower than that of N and Fe."*

We will reword to say: "Large diatoms with thick frustules are known to be prone to rapid sinking, leading to opal dissolution occurring at greater depths, on average, than N and Fe remineralisation. For that reason, a greater proportion of the sinking Si is returned to solution in the deepest box of the model than is N and Fe."

*p. 9-10 "The boundary conditions for Si and N at a specific station are obtained by averaging all available data in a zonal band from 20°E and 120°E, 50° to the east and to the west of the KERFIX longitude (Fig. 5 (a) and 5 (b))." What's the motivation for averaging over such a large range? And why including the area downstream of KERFIX?*

Two reasons:
1) The amount of data is limited. By using a large range, we assure that each latitude is sufficiently represented and that the influence of possible unrepresentative measurements (due to whatever cause) is levelled out.
2) This paper tries to come to general conclusions about the Southern Ocean. From that point of view it is reasonable to use a larger range.

The (deep) boundary conditions for nutrients are not likely to be affected by whether or not the measurements come from downstream of KERFIX (where introduction of iron to the surface stimulates nutrient drawdown). Iron release from Kerguelen island/plateau in any case only affects a small part of the latitudinal range.

*p. 10 "The zonal and temporal dimension of the boundary conditions have therefore no meaning in the model." Do you mean 'zonal and temporal variations have been averaged out'? Although the KERFIX station is located 60 miles southwest of Kerguelen Islands (upstream with respect to ACC & westerlies) one might expect a local iron input. Are there any iron measurements available and what do they tell us?*

Indeed, that is exactly what we mean. We will make this clearer. We use average concentrations in the zonal direction. We do this for the same reason as above: there is not enough data to do otherwise.

*p. 12 "Despite the attractive simplicity of this assumption, it makes comparing model results with one localised sampling dataset (obtained during a specific cruise, or satellite mission, acquired at a certain time in year, or using specific methods, etc.) complex." Instead of 'complex' I would say 'difficult' or even 'impossible'. Between the early 1990ies and the modeling period (2009-2012) the wind forcing (SAM index!) has changed quite a bit!*

We acknowledge that the model runs for the period 2009 – 2012 (when B-SOSE model outputs are available) whereas the KERFIX dataseries dates back to the 1990ies.

We would like to draw attention to the main intention of using KERFIX as a point of comparison for our model:

- It is one of the only (if not the only) datasets where we have information on nutrients and phytoplankton concentrations over a larger time span. This is what makes it a very useful dataset for checking a model. The B-SOSE and GLODAPv2 datasets are used as inputs for our model. The KERFIX timeseries is the only proper dataset that we can use for validation of the model.
- It is argued that this dataset is representative for the Southern Ocean HNLC region
- We fully acknowledge that model results are not entirely in line with the measurements. Perfect agreement is, as the reviewer mentions, not to be expected for several reasons:
  - Difference in time frame
  - Localised versus very averaged model results
  - Hourly model results versus monthly sampling data

The point of comparing model results with KERFIX data is therefore not to completely reproduce the KERFIX dataset, but to demonstrate to the reader that the model generates reasonable results for the purpose intended.

*p. 13 ”The units of the phytoplankton biomass are converted from mmol N m⁻₃ to mg chla m⁻₃.” This is not just a change of units! The different units indicate different measures of biomass.*

Correct. We will adapt to: "The biomass of phytoplankton is expressed in mg chl-$a$ m$^{-3}$…"

*p. 13 units missing: Redfield is in mol mol⁻₁, C:chl is in g g⁻₁*

Correct, we will adjust.

*p. 26 ”Biogeochemical models of, or including, the SO must include the process of entrainment as accurately as possible if they are to hope to reproduce reality.” All biogeochemical general circulation models (BGCMs) include entrainment. Which models do not use entrainment?*

Our point is not that models do not include entrainment at all but rather they need to do it accurately.

*p. 26 "When biology was turned off, while maintaining the deep gradient, the model still reproduced a strong Si gradient." How can you maintain or generate the deep gradient without biology?*

Correct. We do not argue that biology is unimportant, only that it is less important than physics over short timescales. We will alter the text to make this clearer, but also reiterate it here.

Our main conclusion is that physical processes are primarily responsible for much stronger proportional decline in SiO4 than in NO3 over the timescale that surface waters advect northwards across the ACC towards mode water subduction zones. The existence in reality of Si and N gradients at depth is the reason that the model is able to reproduce the pattern in the mixed layer even when biology is turned off. At depth, Si has a stronger gradient than N, as can be seen in the boundary conditions (real data). However, due to the fixed boundary conditions, the model is not useful in explaining why we have that gradient at depth. For this reason, we focus on short timescales.

It is not as simple as 'biology is unimportant' and 'physics is important'. We created a simplified model of a section in the SO and we concluded that:
- The N-gradient as found in the SO meridional sections is a reflection of the N-gradient at depth along that section
- The Si-gradient as found in the SO meridional sections is a reflection of the Si-gradient at depth along that section
- Without physical processes, we do not find a gradient in the mixed layer in the model
- Without a nutrient gradient at depth, we do not find a gradient in the mixed layer in the model

We tried to be unambigious in the timescales for which we make this claim. To answer the specific question of the reviewer: this model works on timescales of a few years. The dynamic interaction between surface and deep water and biology, in which the chemical composition of deep water is allowed to change, plays out on longer timescales (see for instance the paper by Holzer et al, cited in our manuscript). As noted in our response to reviewer 1, it is hard with the model we used to be precise about exactly how long it is before biology becomes dominant. We cannot use our model to address controls over longer timescales because it is not suitable for that purpose.

*p. 31 "Mawji, E. and et al.: The Geotraces Intermediate Data Product 2014, Marine Chemistry, 2015." please complete reference and drop 'and'*

We will do this.

*p. 32 What's the status of: "Verdy, A. and Mazloff, M. R.: A coupled physical-biogeochemical data assimilation model for estimating the Southern Ocean carbon system. Submitted to JGR, 2016." ???*

We will cite the published paper (JGR-Oceans, 122 (9): 6968-6988 (2017)).

---

## Referee Report (RR1)

**Review on Demuynck et al.: Spatial Variations in Silicate-to-Nitrate Ratios in the Southern Ocean ... (revised version 8/2019)**

Demuynck et al. simulate the nutrient concentrations (nitrate and silicic acid, or N and Si for short) in the mixed layer (ML) with a 'box model' that allows for spatial resolution in the meridional direction. They responded to the criticisms of two reviewers by detailed comments and various changes in the text. However, in my opinion the manuscript is not yet ready for publication.

The nutrient concentrations in the ML of the Southern Ocean (SO) show strong meridional gradients, especially for Si which in summer decreases to almost zero at the Antarctic Polar Front. Demuynck et al. asked 'Which processes are generating these gradients?'. This is an important question given the fact that the export (via mode and intermediate waters) of nutrients from the SO has impacts for the productivity of large parts of the world oceans. An understanding of the processes involved in generating these gradient is necessary to predict future changes.

The nutrient gradients are generated by two main processes:
(1) 'Biology' (biological production and export of organic matter): acting mainly in spring/summer and depending on energy (light) and nutrients (including micro-nutrients like iron); grazing can play a role for the start and development of algal blooms as well as for export of organic matter.
(2) 'Physics' (upwelling/mixing): these processes can impose 'deeper (few 100 meters) boundary conditions' on ML nutrient conditions; in contrast to biology they act (with varying strength) all year round.

Demuynck et al. point to the importance of physical processes (in combination with given nutrient gradients in deeper layers) to establishing meridional gradients in the ML, especially in winter. This is a valid point and is worth publishing.

I suggest two further improvements of the manuscript:
(A) Clarification of statements about gradients: winter versus summer (see my detailed comments below)
(B) More detailed analysis of model results. The time series shown in Fig. 10 might be a good starting point. It looks as if 'nothing happens'

soon after 'biology' stops. In spring and summer nutrient concentrations are decreased by 'biology', however, restored from time to time by wind-driven upwelling events. Parallel time series of nutrient concentrations, wind forcing, upwelling, horizontal advection, nutrient uptake, export of organic matter etc. at selected stations (especially for Si at high concentrations/60°S versus low concentrations/50°S) for a single year might allow more insight to what is happening. You could integrate over spring, summer, autumn, winter the contribution of the various processes to the change of ML nutrient concentrations at selected stations.

**Detailed comments:**

Fig. 7c: KERFIX data from the early 1990ies are plotted on a time axis from 2009 to 2012: this is fine, however, should be mentioned in the figure legend.

p. 21: The simulated 'absolute' (???) 'contribution of advection/upwelling, remineralisation, biology, diffusive mixing and entrainment to the nitrate concentration in the mixed layer' at station 18 (the information about the station should be added to the figure legend) is shown in Fig.10.

Fig. 11: You might comment on negative contributions (advection).

p. 23 'The observed gradient of N and Si along a south-to-north section is a smooth mirror image of the gradient observed in the boundary conditions for N and Si. This suggests that, if no boundary condition gradient existed, no gradient would be observed in the ML. Indeed, if the model is run with a fixed boundary condition for N (30 mmol m$^{-3}$) and Si (60 mmol m$^{-3}$) along 5 the section, the winter gradient disappears, and winter concentrations of both N and Si are relatively similar along the section with a change of max 10 mmol m$^{-3}$ for Si and 5 mmol m$^{-3}$ for N along the meridional section (Fig. 12(a)). This strongly suggests that the observed meridional nutrient gradient in the ML results from the deep-water nutrient distribution. Having a nutrient gradient below the SSL is a necessary requirement for a nutrient gradient to occur close to the surface.'
These statements are misleading or wrong. It should be made clear from the beginning that these statements refer to the **winter gradients**. In winter time, the mixed layer nutrient concentration are mainly set by (local) vertical upwelling and not by (local) biology. This finding is no surprise

and could be quantified by analyzing the various fluxes (upwelling, mixing, horizontal advection, biology) contributing to the change of nutrient concentrations in a mixed layer box. If horizontal advection and biology contribute little to this change, it is no surprise that after some time the winter gradient in the ML is similar to the concentrations given at the lower boundary (may there be a gradient or not). It would be interesting to know how long it takes for the ML concentrations to 'relax' to the lower boundary conditions (time scale?) and why horizontal advection has only a small impact despite the large northward Ekman transport.

The statement 'Having a nutrient gradient below the SSL is a necessary requirement for a nutrient gradient to occur close to the surface.' is generally wrong because it does not apply in summer (biology will always generate gradients) and needs quantification.

Fig. 12b: Si gradient with biology: 66 mmol m$^{-3}$ at 64°S versus almost 0 mmol m$^{-3}$ at 40°S, i.e. difference = 66 mmol m$^{-3}$
Si gradient without biology: 54 mmol m$^{-3}$ at 64°S versus 26 mmol m$^{-3}$ at 40°S, i.e. difference = 28 mmol m$^{-3}$
Thus biology contributes more than 50% to the overall gradient and it is the only process that can generate very low Si concentrations (in some regions supported by advection).

p. 23: 'Fig. 12(a) and (b) indicate that there would be no ML nutrient gradient at all without a gradient at depth, and with out the connection between the deep and surface waters.' Again: this applies to the gradient in winter. It is not possible to drive Si concentrations to near zero without biology. If the contribution of horizontal advection to setting the ML nutrient concentrations is small, biology is the only process that can generate ML nutrient concentrations lower than deep boundary values.

---

## Referee Report (RR2)

**Review on Demuynck et al.: Spatial Variations in Silicate-to-Nitrate Ratios in the Southern Ocean Surface Waters are Controlled in the Short Term by Physics Rather Than Biology (version 11/2019)**

Demuynck et al. simulate the nutrient concentrations (nitrate and silicic acid, or N and Si for short) in the mixed layer (ML) with a box model that allows for spatial resolution in the meridional direction. They responded to my criticisms by detailed comments and various changes in the text. The authors mention various limitations of their model (especially the restriction to short time scales and the unresolved processes responsible for the lower boundary conditions) allowing the reader to interpret model results appropriately ('Essentially, all models are wrong, some are useful', George Box).

**Detailed comments:**

p.3 '... for the species Actinocyclus and 3:1 for the species Thalassiosira ...' Actinocyclus is a genus (same for Thalassiosira); if only the genus name is known you may use 'Actinocyclus sp.' for an unknown species of genus Actinocyclus

Figure 3 may give the false impression that all boxes have the same size

p.7 '... because our starting base for advection is flow and not velocity of the water ...' -> '... because our starting base for advection is volume transport and not velocity ...'

p.18 '...8 mumol m-3, which is within the range of measured values (12-27 mumol m-3 ...' 8 is outside the range 12-27

p.22 Negative advection???

p.22 'Fig. (b)' add number

p.23 drop '(?)'

---

## Author Response (AR2)

Response to Editor – dated 30 November 2019

Dear Editor,

One of the 2 reviewers of our paper expressed themselves happy with it. We have addressed the comments of the other reviewer as explained below. We hope that the paper will now be considered ready for publication.

Sincerely,
* * *
**Review on Demuynck et al.: "Spatial Variations in Silicate-to-Nitrate Ratios in the Southern Ocean  Surface Waters are Controlled in the Short Term by Physics Rather Than Biology"**

Demuynck et al. simulate the nutrient concentrations (nitrate and silicic acid, or N and Si for short) in the mixed layer (ML) with a 'box model' that allows for spatial resolution in the meridional direction. They responded to the criticisms of two reviewers by detailed comments and various changes in the text. However, in my opinion the manuscript is not yet ready for publication.

The nutrient concentrations in the ML of the Southern Ocean (SO) show strong meridional gradients, especially for Si which in summer decreases to almost zero at the Antarctic Polar Front. Demuynck et al. asked 'Which processes are generating these gradients?'. This is an important question given the fact that the export (via mode and intermediate waters) of nutrients from the SO has impacts for the productivity of large parts of the world oceans. An understanding of the processes involved in generating these gradient is necessary to predict future changes.

The nutrient gradients are generated by two main processes:
(1) 'Biology' (biological production and export of organic matter): acting mainly in spring/summer and depending on energy (light) and nutrients (including micronutrients like iron); grazing can play a role for the start and development of algal blooms as well as for export of organic matter.
(2) 'Physics' (upwelling/mixing): these processes can impose 'deeper (few 100 meters) boundary conditions' on ML nutrient conditions; in contrast to biology they act (with varying strength) all year round.

Demuynck et al. point to the importance of physical processes (in combination with given nutrient gradients in deeper layers) to establishing meridional gradients in the ML, especially in winter. This is a valid point and is worth publishing.

We thank the reviewer for their review and note their view that the main point of our paper is valid and worth publishing.

*I suggest two further improvements of the manuscript:*
(A) Clarification of statements about gradients: winter versus summer (see my detailed comments below)

Response to Editor – dated 30 November 2019

(B) More detailed analysis of model results. The time series shown in Fig. 10 might be a good starting point. It looks as if 'nothing happens' soon after 'biology' stops. In spring and summer nutrient concentrations are decreased by 'biology', however, restored from time to time by wind-driven upwelling events. Parallel time series of nutrient concentrations, wind forcing, upwelling, horizontal advection, nutrient uptake, export of organic matter etc. at selected stations (especially for Si at high concentrations/60⁰S versus low concentrations/50⁰S) for a single year might allow more insight to what is happening. You could integrate over spring, summer, autumn, winter the contribution of the various processes to the change of ML nutrient concentrations at selected stations.

We do not completely agree with the first point (see below). We have however made alterations in response. We agree with the second point and have made the recommended changes (Fig. 10, panel B and new Fig. 11).

**Detailed comments:**

Fig. 7c: KERFIX data from the early 1990ies are plotted on a time axis from 2009 to 2012: this is fine, however, should be mentioned in the figure legend.

Done.

p. 21: The simulated 'absolute' (???) 'contribution of advection/upwelling, remineralisation, biology, diffusive mixing and entrainment to the nitrate concentration in the mixed layer' at station 18 (the information about the station should be added to the figure legend) is shown in Fig.10.

The station information has been added to the legend of Fig. 10. We have added a second panel to Fig. 10 to give more information on the cumulative contributions of different processes. We now cite Fig. 10 in the main text for this information. [pg 19? Lines 20-29]

Fig. 11: You might comment on negative contributions (advection).

Done [pg 22, lines 8-14]

p. 23 'The observed gradient of N and Si along a south-to-north section is a smooth mirror image of the gradient observed in the boundary conditions for N and Si. This suggests that, if no boundary condition gradient existed, no gradient would be observed in the ML. Indeed, if the model is run with a fixed boundary condition for N (30 mmol m$^{-3}$) and Si (60 mmol m$^{-3}$) along 5 the section, the winter gradient disappears, and winter concentrations of both N and Si are relatively similar along the section with a change of max 10 mmol m$^{-3}$ for Si and 5 mmol m$^{-3}$ for N along the meridional section (Fig. 12(a)). This strongly suggests that the observed meridional nutrient gradient in the ML results from the deep-water nutrient distribution. Having a nutrient gradient below the SSL is a necessary requirement for a nutrient gradient to occur close to the surface.'
These statements are misleading or wrong. It should be made clear from the beginning that these statements refer to the winter gradients. In winter time, the mixed layer nutrient concentration are mainly set by (local) vertical upwelling and not by (local) biology. This finding is no surprise and could be quantified by analyzing the various fluxes (upwelling, mixing, horizontal advection, biology)

contributing to the change of nutrient concentrations in a mixed layer box. If horizontal advection and biology contribute little to this change, it is no surprise that after some time the winter gradient in the ML is similar to the concentrations given at the lower boundary (may there be a gradient or not). It would be interesting to know how long it takes for the ML concentrations to 'relax' to the lower boundary conditions (time scale?) and why horizontal advection has only a small impact despite the large northward Ekman transport. The statement 'Having a nutrient gradient below the SSL is a necessary requirement for a nutrient gradient to occur close to the surface.' is generally wrong because it does not apply in summer (biology will always generate gradients) and needs quantification.

Before answering this point of the reviewer, please note that we found a small error in our calculation of the "no biology" run. Another student has been re-running the model to check that we can reproduce all of the results. He could not do so for the "no biology" run which we eventually tracked down to a small mistake in the original code. We have fixed this error and in so doing have resolved the discrepancy. The difference is shown in the figure below. The difference is relatively small and in fact is in the direction of strengthening our main point about the primacy of the physical effect. We have changed the "no biology" lines for winter Si and N in Fig. 13 (b) of the MS and it can be seen the wintertime Si gradient persists without biology, and in fact is less affected by biology than originally calculated. We apologise for the error.

[Figure]

**Figure: winter Si gradient without biology – blue: correct results. Orange: wrong results**

In response to the reviewer, we have altered the latter statement to read *"Having a nutrient gradient below the SSL is a necessary requirement for a nutrient gradient to occur close to the surface. This is especially true in winter, when mixed layers are deepest (Dong et al., 2008) and when mode waters form (Cerovečki et al., 2013)."* [pg 32, lines 18-19]. We agree with the reviewer (see above) that: "*This is an important question given the fact that the export (via mode and intermediate waters) of nutrients from the SO has impacts for the productivity of large parts of the world oceans.*" Because the main aim of our paper is to understand the processes causing mode waters to be low in Si but high in N, our primary focus in on controls on wintertime nutrient distributions rather than summertime ones. Some of the reviewer's comments on this are misconceived for that reason.

However, we appreciate that there is also interest in controls over summertime nutrient distributions. We also realise that we were not sufficiently clear before about our aims. We have made changes to differentiate between summertime and wintertime impacts [Fig. 10 (b) - pg 21, pg 19 lines 20-29 & Fig. 12 - pg 25, pg 22 lines 22 – pg 23 line 8]. We have added a quantification of the contributions of the major fluxes to ML silicate at 40$^o$S [Fig. 12].

Fig. 12b: Si gradient with biology: 66 mmol m$^{-3}$ at 64$^o$S versus almost 0 mmol m$^{-3}$ at 40$^o$S, i.e. difference = 66 mmol m$^{-3}$
Si gradient without biology: 54 mmol m$^{-3}$ at 64$^o$S versus 26 mmol m$^{-3}$ at 40$^o$S, ie. Difference = 28 mmol m$^{-3}$
Thus biology contributes more than 50% to the overall gradient and it is the only process that can generate very low Si concentrations (in some regions supported by advection).

Biology is not the only process that can generate very low Si concentrations. The subsurface concentration (lower boundary condition) for silicate is < 5 mmol m$^{-3}$ at 40$^o$S (Fig. 5). Both wintertime entrainment and biological removal promote low silicate concentrations at 40$^o$S, whereas northwards advection promotes higher values. Horizontal advection always brings more silicate-rich water from further south; it never acts to lower silicate concentrations. These competing influences are illustrated more clearly in the new version of the MS with the added figures [extra panel to Fig. 10 and new Fig. 12].

p. 23: 'Fig. 12(a) and (b) indicate that there would be no ML nutrient gradient at all without a gradient at depth, and with out the connection between the deep and surface waters.' Again: this applies to the gradient in winter. It is not possible to drive Si concentrations to near zero without biology. If the contribution of horizontal advection to setting the ML nutrient concentrations is small, biology is the only process that can generate ML nutrient concentrations lower than deep boundary values.

This is not quite correct – see response to previous point. It is not necessary to find a process that can drive ML values to lower than deep boundary values because deep boundary values for Si are already very low (at 40$^o$S).

[revised manuscript text omitted]

---

## Author Response (AR3)

Response to Editor - dated 16 February 2020

Dear Editor,

We are happy that the paper is considered ready for publication. We would like to thank you for your time and effort. Please find below the small adaptations as suggested by anonymous referee #2.

Sincerely,

**Review on Demuynck et al.: "Spatial Variations in Silicate-to-Nitrate Ratios in the Southern Ocean Surface Waters are Controlled in the Short Term by Physics Rather Than Biology"**

Demuynck et al. simulate the nutrient concentrations (nitrate and silicic acid, or N and Si for short) in the mixed layer (ML) with a box model that allows for spatial resolution in the meridional direction. They responded to my criticisms by detailed comments and various changes in the text. The authors mention various limitations of their model (especially the restriction to short time scales and the unresolved processes responsible for the lower boundary conditions) allowing the reader to interpret model results appropriately ('Essentially, all models are wrong, some are useful', George Box).

**Detailed comments:**

p.3 '... for the species Actinocyclus and 3:1 for the species Thalassiosira ...' Actinocyclus is a genus (same for Thalassiosira); if only the genus name is known you may use 'Actinocyclus sp.' for an unknown species of genus Actinocyclus

Ok, done (p.3 - line 7)

Figure 3 may give the false impression that all boxes have the same size

Please see p.5 line 10. We have added an extra line to table 1 showing the location of each station.

p.7 '... because our starting base for advection is flow and not velocity of the water ...' -> '... because our starting base for advection is volume transport and not velocity ...'

Ok, done (p.7 - line 5)

p.18 '...8 mumol m-3, which is within the range of measured values (12-27 mumol m-3 ...' 8 is outside the range 12-27

We now say: " which is close to the range of measured values … " (p.17 - line 16)

p.22 Negative advection???

We now say: "negative contribution of advection" (p.22 - line 13)

p.22 'Fig. (b)' add number

Ok, done (p.22 - line 30)

p.23 drop '(?)'

Ok, done

**Spatial Variations in Silicate-to-Nitrate Ratios in the Southern Ocean Surface Waters are Controlled in the Short Term by Physics Rather Than Biology**

Pieter Demuynck[1], Toby Tyrrell[1], Alberto Naveira Garabato[1], Mark C. Moore[1], and Adrian P. Martin[2]

[1]Ocean and Earth Science, University of Southampton, Southampton SO14 3ZH, UK
[2]National Oceanography Centre, Southampton SO14 3ZH, UK

**Correspondence:** Toby Tyrrell (toby.tyrrell@soton.ac.uk)

**Abstract.** The nutrient composition (high in nitrate but low in silicate) of Subantarctic Mode Water (SAMW) forces diatom scarcity across much of the global surface ocean. This is because diatoms cannot grow without silicate. After formation and downwelling at the Southern Ocean's northern edge, SAMW re-emerges into the surface layers of the mid- and low-latitude oceans, providing a major nutrient source to primary producers in those regions. The distinctive nutrient composition of SAMW originates in the surface waters of the Southern Ocean, from which SAMW is formed. These waters are observed to transition from being rich in both silicate and nitrate in high-latitude areas of the Southern Ocean, to being nitrate-rich but silicate-depleted in SAMW formation sites further north. Here we investigate the key controls of this change in nutrient composition with an idealised model, consisting of a chain of boxes linked by a residual (Ekman- and eddy-induced) overturning circulation. Biological processes are modelled on the basis of seasonal plankton bloom dynamics, and physical processes are modelled using a synthesis of outputs from the data-assimilative Southern Ocean State Estimate. Thus, as surface water flows northward across the Southern Ocean toward sites of SAMW formation, it is exposed in the model (as in reality) to seasonal cycles of both biology and physics. Our results challenge previous characterisations of the abrupt northward reduction in silicate-to-nitrate ratios in Southern Ocean surface waters as being predominantly driven by biological processes. Instead, our model indicates that, over shorter timescales (years to decades), physical processes connecting the deep and surface waters of the Southern Ocean (i.e. upwelling and entrainment) exert the primary control on the spatial distribution of surface nutrient ratios.

*Copyright statement.* The works published in this journal are distributed under the Creative Commons Attribution 4.0 License. This licence does not affect the Crown copyright work, which is re-usable under the Open Government Licence (OGL). The Creative Commons Attribution 4.0 License and the OGL are interoperable and do not conflict with, reduce or limit each other. ©Crown copyright 2019

[revised manuscript text omitted]